# Impact of cervical screening by human papillomavirus genotype: Population-based estimations

**Jiangrong Wang**[1], **K. Miriam Elfström**[1,2], **Camilla Lagheden**[1], **Carina Eklund**[1], **Karin Sundström**[1], **Pär Sparén**[3], **Joakim Dillner**[1,2]*

1 Division of Cervical Cancer Elimination, Department of Clinical Science, Intervention and Technology, Karolinska Institutet, Stockholm, Sweden, 2 Department of Pathology and Cancer Diagnostics, Medical Diagnostics Karolinska, Karolinska University Hospital, Stockholm, Sweden, 3 Department of Medical Epidemiology and Biostatistics, Karolinska Institutet, Stockholm, Sweden

* joakim.dillner@ki.se

**Data Availability Statement:** Relevant data files other than published as figures and tables in the manuscript is stored in B2SHARE: https://doi.org/

## Abstract

### Background

Cervical screening programs use testing for human papillomavirus (HPV) genotypes. Different HPV types differ greatly in prevalence and oncogenicity. We estimated the impact of cervical screening and follow-up for each HPV type.

### Methods and findings

For each type of HPV, we calculated the number of women needed to screen (NNS) and number of women needing follow-up (NNF) to detect or prevent one cervical cancer case, using the following individual level input data (i) screening and cancer data for all women aged 25 to 80 years, resident in Sweden during 2004 to 2011 (N = 3,568,938); (ii) HPV type-specific prevalences and screening histories among women with cervical cancer in Sweden in 2002 to 2011(N = 4,254); (iii) HPV 16/18/other HPV prevalences in the population-based HPV screening program (N = 656,607); and (iv) exact HPV genotyping in a population-based cohort (n = 12,527). Historical screening attendance was associated with a 72% reduction of cervical cancer incidence caused by HPV16 (71.6%, 95% confidence interval (CI) [69.1%, 73.9%]) and a 54% reduction of cancer caused by HPV18 (53.8%, 95% CI [40.6%, 63.1%]). One case of HPV16-caused cervical cancer could be prevented for every 5,527 women attending screening (number needed to screen, NNS). Prevention of one case of HPV16-caused cervical cancer required follow-up of 147 HPV16–positive women (number needed to follow-up, NNF). The NNS and NNF were up to 40 to 500 times higher for HPV types commonly screened for with lower oncogenic potential (HPV35,39,51,56,59, 66,68). For women below 30 years of age, NNS and NNF for HPV16 were 4,747 and 289, respectively, but >220,000 and >16,000 for HPV35,39,51,56,59,66,68. All estimates were either age-standardized or age-stratified. The primary limitation of our study is that NNS is dependent on the HPV prevalence that can differ between populations and over time. However, it can readily be recalculated in other settings and monitored when HPV type-specific

10.23728/b2share.
ffe80b8d159441a7b818070cb8cc5482.

**Funding:** JD received funding from the European Union's Horizon 2020 Research and Innovation Programme under grant agreement No.847845 (Project RISCC) https://research-and-innovation.ec.europa.eu/funding/funding-opportunities/funding-programmes-and-open-calls/horizon-2020_en JD also received funding from the Swedish Cancer Society, with project number 20 1198 PjF 01 H and 20 1199 UsF 02 H https://www.cancerfonden.se/ The funders had no role in study design, data collection and analysis, decision to publish, or preparation of the manuscript.

**Competing interests:** KS has received research grants from Merck and Co, LLC, for register-based follow-up of HPV vaccination in Sweden. This interest has no financial stake in the result of the current study. Other authors have declared that no competing interests exist.

**Abbreviations:** CI, confidence interval; HPV, human papillomavirus; IARC, International Agency for Research on Cancer; NNF, number needed to follow-up; NNS, number needed to screen; WHO, World Health Organization.

prevalence changes. Other limitations include that in some age groups, there was little data and extrapolations had to be made. Finally, there were very few cervical cancer cases associated with certain HPV types in young age group.

## Conclusions

In this study, we observed that the impact of cervical cancer screening varies depending on the HPV type screened for. Estimating and monitoring the impact of screening by HPV type can facilitate the design of effective and efficient HPV-based cervical screening programs.

## Trial registration

ClinicalTrials.gov with numbers NCT00479375, NCT01511328.

## Author summary

### Why was this study done?

- Cervical screening programs now use testing for human papillomavirus (HPV).

- Different HPV types differ greatly in prevalence and oncogenicity, therefore screening for and further management of certain HPV types may cause excessive false positives and resource consumption.

- How cervical screening program may be impacted by screening for different HPV types has not been sufficiently studied.

### What did the researchers do and find?

- We integrated the Swedish nationwide data of HPV genotype and cervical screening history among cervical cancer cases as well as the general population and calculated "number needed to screen" and "number needing follow-up" for preventing and detecting one case of cervical cancer caused by each HPV type.

- The impact of cervical screening was very different from different HPV types: prevention or detection of one cervical cancer case caused by HPV16 involved much fewer women in screening and required much fewer being followed up, as compared to types with lower oncogenic potential, such as HPV35, 39, 51, 56, 59, 66, 68.

- In young women, screening and follow-up of HPV35, 39, 51, 56, 59, 66, 68 would require unreasonably large efforts per prevented or detected case, whereas in older women, screening and follow-up of these HPV types appeared reasonable.

- HPV18-related cervical cancer was inadequately prevented in cytology-based screening.

### What do these findings mean?

- Cervical screening programs may consider selecting which HPV types to screen for or follow-up, depending on women's age.

- HPV vaccination is changing the HPV type-specific prevalence in the population, thus monitoring the impact of screening by HPV type can facilitate the design of effective and efficient HPV-based cervical screening programs.

- The major limitation is that HPV prevalences are changing over time, necessitating updated calculations of the impact.

## Introduction

Elimination of cervical cancer as a public health problem is a globally prioritized goal issued by the World Health Organization (WHO). To achieve this goal, screening efforts that use an optimal screening test and management algorithms are needed. Following results from randomized clinical trials that demonstrated a greater cancer-protective effect when screening using human papillomavirus (HPV) testing as compared to cytology [1], HPV-based screening is now the globally recommended screening strategy [2].

Cervical cancer is caused by infection with oncogenic HPV types, of which the International Agency for Research on Cancer (IARC) recognizes 12 HPV types as oncogenic (HPV16, 18, 31, 33, 35, 39, 45, 51, 52, 56, 58, and 59) and 1 HPV type as "probably oncogenic" (HPV68) [3]. It is well established that the HPV type-specific cervical cancer risks vary greatly across HPV types [4,5], with the most oncogenic HPV type (HPV16) associated with a >20-fold increased cancer risk and the least oncogenic HPV type (HPV51) associated with a risk increase of only about 1.2-fold [5]. As some of the HPV types with limited oncogenicity are also common infections in the population [6], screening for these types and managing all women positive for these types would consume large amount of resources that may impair handling higher risk groups, as resources are never unlimited, and may result in overtreatment especially in young women.

Today, there are several HPV testing platforms available that can provide extended HPV genotyping in screening [7]. Utilizing this information in screening is under increasing discussion, and current evidence suggests its value on risk discrimination for resource allocation [8–11]. To design an efficient and effective HPV-based cervical screening program, knowledge only of oncogenicities in relative risk and prevalences of different HPV types is not enough, knowledge about screening resources and follow-up resources required to achieve benefit is also needed. Hence, an intuitive measurement integrating information of prevalence, oncogenicity, and screening effectiveness of each HPV type, meanwhile accomodating resource-benefit quantification, should be highly informative. Impact numbers [12] are suitable measurement. Population and disease impact numbers, developed from "number needed to treat statistic," are defined as "the number of those in the whole population among whom one event will be prevented by the intervention," and "the number of those with the disease in question among whom one event will be prevented by the intervention," respectively [12]. To the best of our knowledge, there has not been any report of impact numbers or similar assessment of the efficiency of cervical screening program by HPV type.

Sweden has the infrastructure to obtain the data needed to calculate the impact numbers, as screening is based on an organized, high-coverage cervical screening program closely following WHO/IARC recommendations (S1 Appendix), and comprehensive individual-level data on HPV testing, cervical screening, and cervical cancer is collected in registries and population-based randomized trials. For each one of the 12 oncogenic HPV types, and for 2

additional HPV types commonly included in HPV tests (66 and 68, classified as possibly and probably oncogenic, respectively, by WHO/IARC [3]), we integrated the population prevalence, oncogenicity, and cancer prevention potential to quantify the population level impact number for screening: number of women in the screening target population among whom one cervical cancer case caused by a certain HPV type can be prevented (number need to screen, NNS), as well as disease impact number: number of screen–positive women for an HPV type who need follow-up to prevent one cancer case (number needing follow-up, NNF). As cervical screening also aims to reduce mortality by early detection [13], and not all cervical cancer can be prevented even with adequate screening, we also calculated the corresponding impact numbers for screen-detection of one residual cancer case that is not prevented by screening. These impact numbers aim to elucidate the efficiency of HPV genotyping in cervical screening and inform decision-making of which HPV types to screen for and manage.

## Materials and methods

This study is reported as per the Strengthening the Reporting of Observational Studies in Epidemiology (STROBE) guideline (S1 STROBE Checklist).

### Study population and data sources

This study was based on the overall population of women living in Sweden since the 1990s. We retrieved and integrated individual-level data from a variety of data sources to generate 3 key parameters serving for the impact number calculation. The 3 indicators are (i) incidence of invasive cervical cancer in the population by screening history; (ii) HPV type-specific prevalence in the population; and (iii) HPV type distribution among cervical cancer cases by screening history (Fig 1).

Incidence rate of invasive cervical cancer in the population, the average from 2004 to 2011 by screening history in the 10 years prior to each calendar year, was assessed in women aged between 25 and 80 years, i.e., ages at which cervical cancer that can potentially be prevented through screening according to the screening program in Sweden (S1 Appendix). This was calculated through individual-level data linkage across the Swedish Total Population Registry, the Swedish National Cancer Registry [14], and the Swedish National Cervical Screening Registry (NKCx [15,16], S1 Appendix) (protocol included in S1 Protocol). Cytology-based screening was performed in the historical period.

HPV type-specific prevalences in the population were retrieved from NKCx among >390,000 women in the capital region of Stockholm, as well as the Swedescreen population-based randomized clinical trial of HPV-primary screening which enrolled 12,527 women from 5 major cities in Sweden. The capital region of Stockhom represents 20% of the Swedish population, and during the early years of HPV-based screening in Sweden, about 80% of all HPV tests were performed at the central HPV testing laboratory at the Karolinska University Laboratory in Stockholm. Therefore, restricting the data to this region ensured that all tests (among women ages 30 to 64 years who were participating in organized HPV-primary screening during 2012 to 2019) had been performed by exactly the same protocol (the Roche Cobas 4800 platform that tests for HPV16, 18 and a combination of 12 "other" oncogenic HPV types). The systematic implementation and evaluation of the primary HPV screening in this region has been well characterized in previous papers [17,18] (protocol included in S2 Protocol). To complement the data with results for women aged 23 to 29 years (who during the study period of 2012 to 2019 were tested with primary cytology), using exactly the same HPV testing platform, we retrieved 592 archival cervical screening samples from the Stockholm Cervical Cytology Biobank. The biobank systematically stores all cervical screening samples in the Stockholm

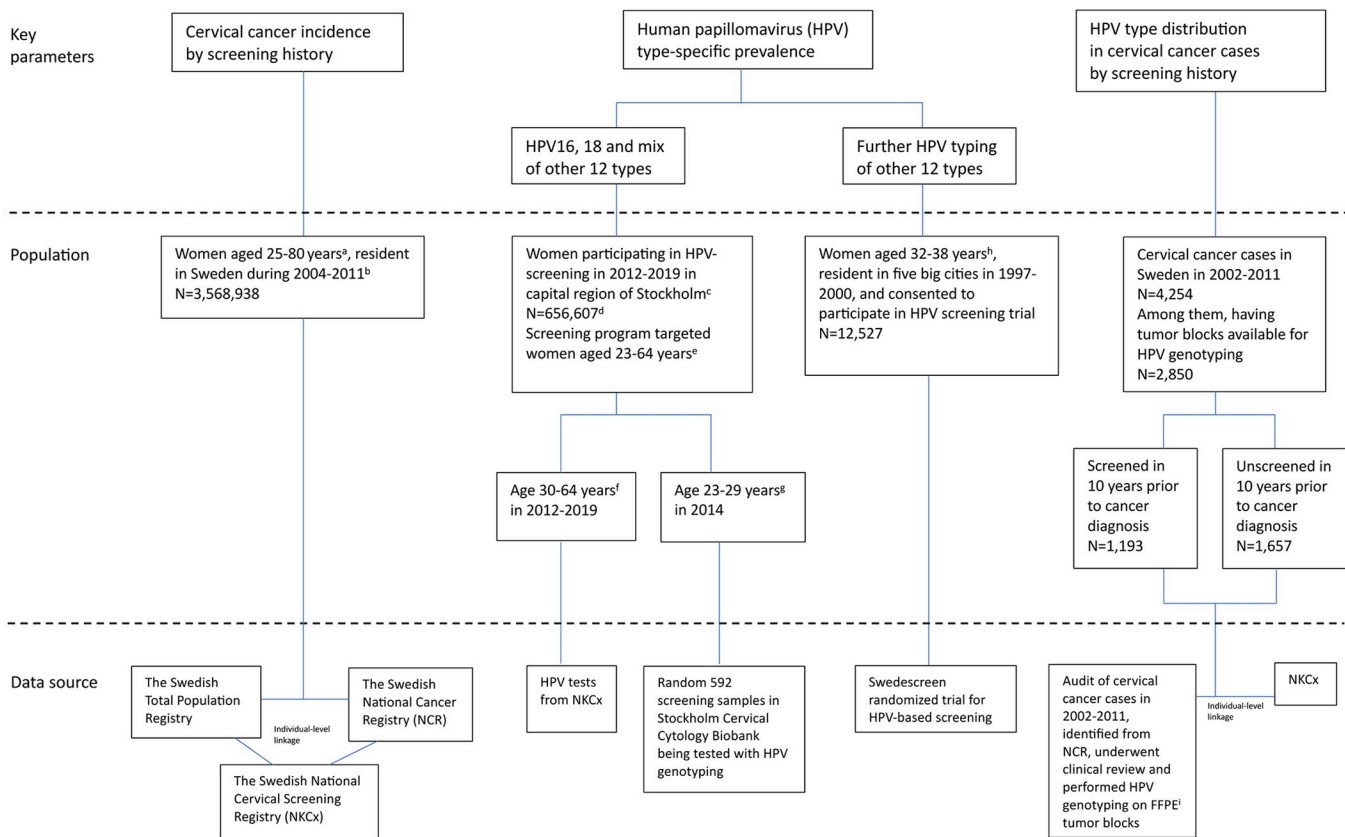

**Fig 1. Flowchart of study population and data resources.** (a) Ages of cervical cancer that can potentially be prevented by screening [38,39]. (b) Starting year 2004 was to allow 10-year screening history from the screening registry NKCx that reached full coverage during 1993–1995. Ending year of 2011 was to adapt to the Audit project with HPV genotyping in cervical cancer cases. (c) The capital region of Stockholm contains 20% of the entire Swedish population. In 2012–2016, the capital region of Stockholm initiated the healthcare policy trial randomizing half of the screening population to HPV-primary screening [17,18]. It was the only region implemented population-based HPV-primary screening at that time. From 2017, the capital region of Stockholm implemented HPV-primary screening for all women aged 30–64, using Cobas platform for HPV partial typing. In 2018–2020, other regions of Sweden gradually implemented HPV-primary screening in the population. (d) Corresponding population of women eligible for cervical screening during the same years in capital region of Stockholm: N = 848,211. (e) Ages eligible for cervical screening (S1 Appendix). (f) Ages eligible for HPV-primary screening at beginning. (g) Ages not recommended for HPV-primary screening at beginning. (h) Age eligible for Swedescreen randomized trial of HPV-based screening [19]. (i) FFPE: formalin-fixed paraffin-embedded. HPV, human papillomavirus.

region. Finally, to estimate the prevalence of specific HPV types contained in the mix of "other" 12 HPV types, we used the age-specific prevalence of the mixed "other" HPV types from NKCx, adding on the composition of each type from the Swedescreen population-based randomized clinical trial that enrolled 12,527 women aged 32 to 38 years participating in organized cervical screening in 5 major cities in Sweden during 1997 to 2000 and performed HPV genotyping on all HPV–positive samples [19–21] (protocol available at https://clinicaltrials.gov/study/NCT00479375).

The estimation of the HPV type-specific prevalence in the general population was based on the following conditions and assumptions. First, the screening population was considered to be representative of the entire population. As >80% of women in Sweden participate in screening according to recommendations, and 92% have at least 1 sample on record in a 10-year period [15], we considered that the HPV type-specific prevalence in the screening population could largely represent that in the general population. As the exact composition of oncogenic HPV types other than 16 and 18 are only available for the Swedescreen participants who were in the ages 32 to 38 years and sampled during 1997 to 2000, we assumed that the

relative composition of the specific HPV types among the "other" positives had not changed substantially since the trial was performed and that the relative distribution among the "other" HPV types was not substantially different by age. Data were sparse comparing the composition of HPV types across ages in the literature. We found 2 publication suggesting that the distribution of high-risk HPV types other than 16 and 18 is roughly proportional across age groups [22,23], so our extrapolation should be acceptable. For calendar period difference, we compared the prevalence of HPV16 and 18 between NKCx in 2012 to 2019 and Swedescreen in 1990s and found them comparable (among women in their 30s, HPV16 prevalance was 2.2% and 2.3%, respectively; HPV18 was 0.7% and 0.5%, respectively).

The HPV type distribution among cervical cancer cases in Sweden was retrieved from the Swedish National Audit of Cervical Cancer Cases in 2002 to 2011. It was assessed by first establishing a list of all 4,254 cervical cancer cases in Sweden during 2002 to 2011 from the Swedish National Cancer Registry and then requesting the archival diagnostic tissue block from the respective pathology departments in Sweden. Overall, tumor blocks could be retrieved and HPV genotyped for 2,850 cases. HPV genotyping was completed using polymerase chain reaction and was complemented with whole-genome sequencing [24,25]. Through individual-level data linkage with NKCx, we presented HPV type distribution among cervical cancer cases who were screened and unscreened in the 10 years prior to cancer diagnosis (protocol included in S1 Protocol). Cytology-based screening was performed in the historical period.

The above 3 parameters were based on data from varied calendar periods from 1990s to 2019 due to availabilities of different data sources, as explained in legend of Fig 1. We assumed that the HPV genotype distribution did not change substantially over these 20 to 30 years, which was to a certain extent supported by the aforementioned finding that the population prevalence of HPV16 and 18 in 2012 to 2019 was comparable to that in 1997 to 2000. HPV vaccination has not yet noticeably affected the study population included in this study: no data was from birth-cohorts of women being vaccinated in the school-based or similar high-coverage HPV vaccination program. No selection of the data was made on ethnicity or other factors.

## Statistical analysis

We calculated and plotted the age-specific incidence rate of invasive cervical cancer in Sweden during 2004 to 2011, by screening history within the 10 years preceding each calendar year. We also plotted the age-specific prevalence of HPV16, 18, and "other" 12 types (31, 33, 35, 39, 45, 51, 52, 56, 58, 59, 66, and 68) in the Stockholm screening population during 2012 to 2019. We further tabulated the percentage of 14 major HPV types among the Swedish cervical cancer cases during 2002 to 2011 by screening history in the 10 years preceding diagnosis. Tests within 6 months prior to the cancer diagnosis were not considered to be the tests with potential to prevent the cervical cancer, but rather part of the diagnostic procedure [26]. Hence, the time period defining the screened and unscreened cases was the 10 to 0.5 years prior to cervical cancer diagnosis.

We used the age-standardized cervical cancer incidence in the population by screening history and HPV type distribution of cases by screening history to estimate number of cervical cancer cases with each HPV type in the pseudo-scenarios (i) if all women were screened; and (ii) if all women were unscreened in the preceding 10 years. The scenarios were compared and the number and percentage of cases that were preventable through screening, by HPV type, was calculated. This estimation is based on the assumption that the screened population would have had the same risk of cervical cancer as the unscreened population should they be unscreened. This assumption should largely hold in the Swedish setting, according to our

previous study which showed that the differences of cervical cancer incidence between screened and unscreened group was dominated by the screening participation itself and not confounded by factors of education and country of birth of individuals [26].

We presented the age-standardized population prevalence of the 14 major HPV types, as well as their risk profiles calculated as number of invasive cervical cancer cases of each HPV type per 1,000 women positive for the type in the unscreened scenario, in a 2D graph.

For each HPV type, we calculated the number of women in the target population among whom one cervical cancer case caused by each HPV type is prevented or detected (number needed to screen, NNS) and the number needed to follow-up of women positive for certain HPV type (NNF) to prevent or detect one cervical cancer case (protocol follows [12]). This was done using the age-standardized percentage of each HPV type among screened and unscreened cases in the last 10 years, as well as the age-standardized population prevalence of each HPV type. The NNS to prevent one case was calculated as total number of women in the population in a year ($N_{population}$) divided by the difference between the number of cancer cases ($N_{case}$) with a particular HPV type (type X) in a year in the pseudo-scenario that all women were unscreened and the number of cases with that type in a year in the pseudo-scenario that all women were screened:

$$NNS_{prevent} = \frac{N_{population}}{(N_{case,typeX}|All\ unscreened) - (N_{case,typeX}|All\ screened)}$$

The NNS to detect one case was calculated as total number of women in the population in a year divided by number of cancer cases with a particular HPV type in a year, in the pseudo-scenario that all women were screened:

$$NNS_{detect} = \frac{N_{population}}{(N_{case,typeX}|All\ screened)}$$

NNF to prevent one case was calculated as number of women who tested positive for a particular HPV type in a year ($N_{typeX+}$) divided by the difference between the number of cancer cases with a particular HPV type in a year, in the pseudo-scenario that all women were unscreened and the number of cases with that type in a year in the pseudo-scenario that all women were screened:

$$NNF_{prevent} = \frac{N_{typeX+}}{(N_{case,typeX}|All\ unscreened) - (N_{case,typeX}|All\ screened)}$$

In the pseudo-scenario that all women were screened, the NNF to detect one case was calculated as number of women who tested positive for a particular HPV type in a year divided by number of cancer cases with a particular HPV type in a year:

$$NNF_{detect} = \frac{N_{typeX+}}{(N_{case,typeX}|All\ screened)}$$

The confidence intervals (CIs) for the percentage of preventable cases as well as the impact numbers by HPV types were estimated through bootstrap resampling [27] of 2,850 cervical cancer cases with HPV genotyping. We resampled the 2,850 cases 1,000 times with replacement and presented the 25th and 975th values of ranked percentage of preventable cases and impact numbers, as lower and upper confidence limit.

Age standardization was performed to (i) control for the different age distribution in the screened and unscreened population in general; and (ii) to report the overall impact numbers

not limiting to populations with the same age structure. Impact numbers were further reported with age-stratification. We kept the estimation simplified with age as the only controlled factor, because (i) the aim is to present impact numbers that can be refered to or reproduced in other settings where factors other than age may not be available; and (ii) according to our previous research, no other demographic or socioeconomic factors had noticeably biased the effect of screening on cervical cancer prevention [26].

Due to sparse number of cancer cases for certain HPV types, in certain result reports we grouped HPV 31, 33, 52, and 58 as intermediate oncogenic types, and HPV 35, 39, 51, 56, 59, 66, and 68 as lower oncogenic types (etiological fraction less than 2%). This is based on the evaluation by WHO/IARC [4].

All data management and analyses were performed in SAS 9.4.

### Ethical statement

The analysis using data linkage between NKCx and cancer registry was approved by Ethical Review Board in Stockholm, Sweden with decision number DNR 02–556. The HPV genotyping of cervical cancer cases in 2002 to 2011 was approved by Ethical Review Board in Stockholm, Sweden with decision number DNR 2011/1026-31/4 and DNR 2012/1028-32. The Swedescreen study was approved by the Ethical Review Board in Stockholm, Sweden with decision number DNR 1996/305 and DNR 2012/780-32. In Sweden, ethical permissions are given by a government agency (The Swedish Ethical Review Agency) that is chaired by a senior judge and has the authority to decide on the formats for information and consent. For the data from the national Swedish cervical screening registry, the decision was that consent was not required and for the trials of HPV testing in cervical screening, the decision was verbal consent after having received written information.

### Results

The incidence of invasive cervical cancer in Sweden during 2004 to 2011 was found to be considerably higher among women unscreened than screened in last 10 years (Fig 2). The overall age-standardized incidence rate in the population, average over 8 years and based on Swedish population in 2000, was 9.9 per 100,000 person-years, and the age-standardized incidence rate of the screened and unscreened groups were 8.4 and 25.6 per 100,000 person-years, respectively (Fig 2).

The prevalence of oncogenic HPV in the population was strongly dependent on age. Among >390,000 women in the capital region of Stockholm, close to 30% of the population were positive for HPV in ages 23 to 29 years, but only 6% to 7% were HPV positive after 50 years of age (Fig 3). The age-standardized population prevalence of 14 HPV types varied from 0.18% (HPV68) to 2.67% (HPV16) (Fig 4). Among cervical cancer cases, more HPV16, less HPV18 and more "other" HPV types were found in previously unscreened cases as compared to screened cases, and younger cases were related to fewer types of HPV (Table 1). Fig 4 compares the prevalence and risk profile across 14 HPV types in the pseudo-scenario of no screening in a 2D graph. The y-axis displays the age-standardized prevalence in percentage of each type in the population, and the x-axis displays the incidence of cervical cancer among women positive for each HPV type in absence of screening. HPV16, at the top-right corner of the graph, had both high prevalence and high risk in the population, whereas HPV59, 66, and 68 at the bottom-left corner, had both low prevalence and low risk. Certain HPV types had low prevalence and high risk, for example, HPV18 and 33, and certain types had high prevalence and low risk, for example, HPV51.

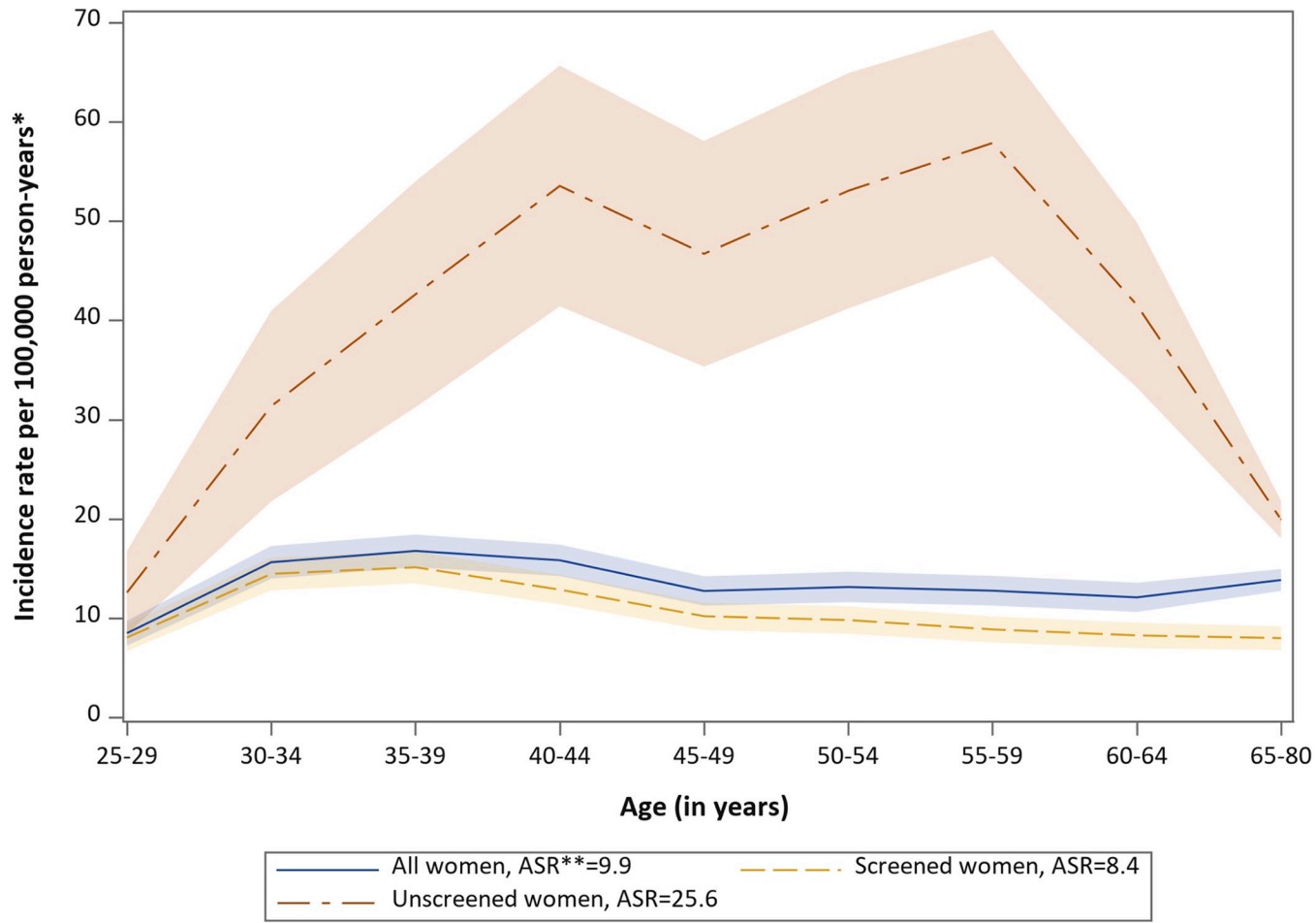

**Fig 2. Age-specific incidence rate of invasive cervical cancer in Sweden 2004–2011 by screening status within 10 years prior to each calendar year.** *
Average from 2004 to 2011 by screening history in the 10 years prior to each calendar year. **ASR = Age-standardized rate.

Screening likely contributed to approximately 72% (95% CI [69%, 74%]) reduction in cases caused by HPV16 and by types other than 16 and 18, whereas it contributed to only 54% (95% CI [41%, 63%]) reduction in cases caused by HPV18 (Table 2).

The impact numbers for cervical screening in the population, i.e., NNS and NNF to detect or prevent one case of cervical cancer, varied substantially among the 14 HPV types investigated. Taking HPV16 as an example, one cervical cancer case caused by HPV16 would be prevented among every 5,527 women in the screening program (95% CI [5,076 to 6,054]), and among women tested positive for HPV16, performing clinical follow-up of 147 HPV16–positive women would prevent one case (95% CI [135 to 161]). For HPV59, the corresponding numbers were 1 case per every 1,339,680 women in screening (95% CI [404,361 to infinity]) and follow-up of 4,389 HPV59–positive women to prevent one case (95% CI [1,324 to infinity]). For HPV51, NNS and NNF could not be estimated, because too few HPV51–positive cervical cancer cases were detected during 10 years in the country of Sweden (Table 3 and Fig 5 and in S1A Table).

The impact numbers by HPV types varied by age. Among women aged below 30 years, the NNS and NNF to prevent one cervical cancer case by HPV16 was about 50 to 60 times lower as compared to the low oncogenicity group of viruses (HPV35, 39, 51, 56, 59, 66, 68), whereas

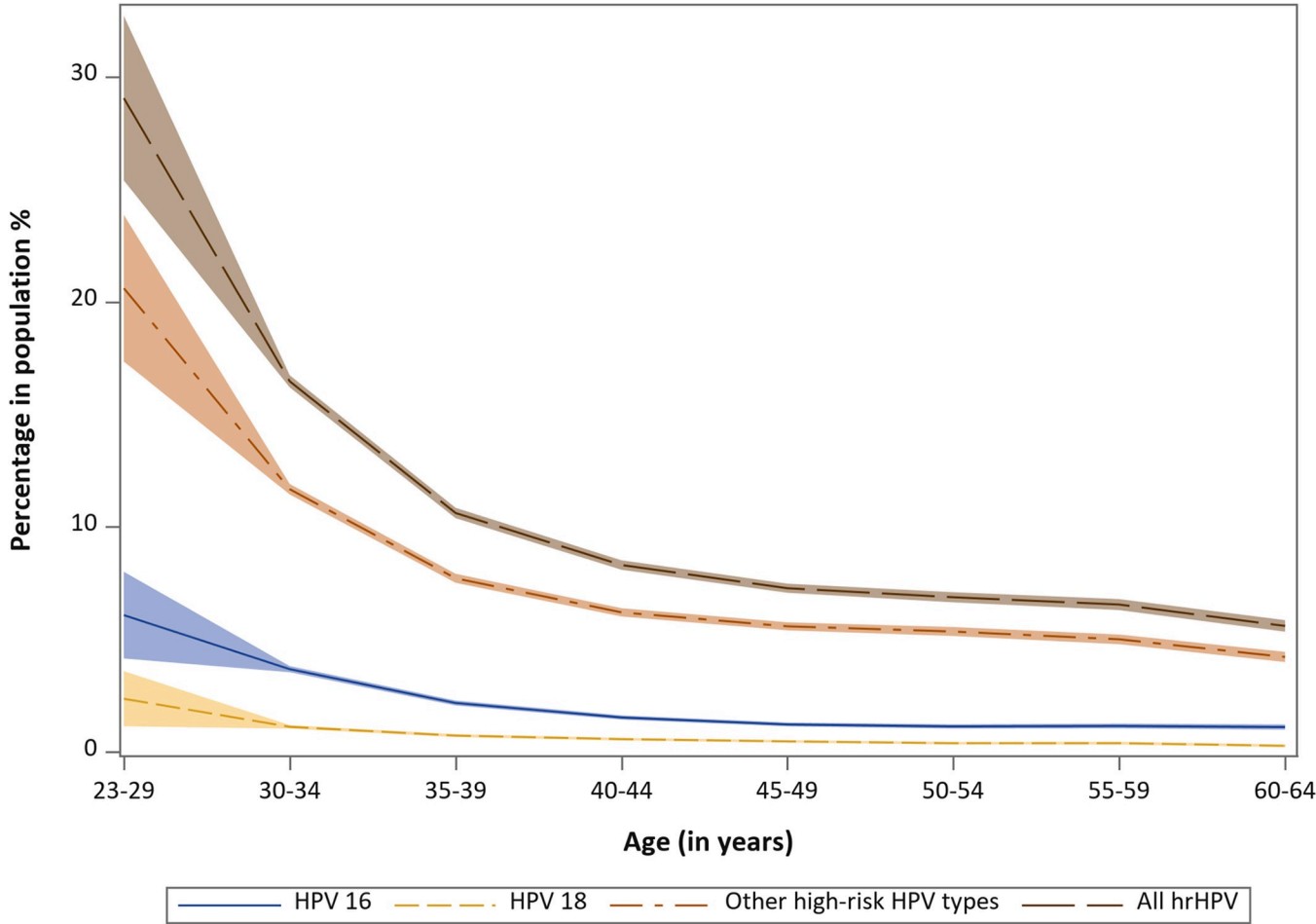

**Fig 3. Percentage positivity with 95% CIs of HPV16, 18, and 12 "other" HPV types by age among >390,000 women participating in organised cervical screening in the capital region of Sweden.** CI, confidence interval; HPV, human papillomavirus.

among women aged between 51 and 60 years, the NNS and NNF to prevent one cervical cancer case by HPV16 was only about 10 to 20 times lower compared to the low oncogenicity group of viruses (Table 4 and Fig 6 and S1B Table).

## Discussion

We found that the 12 oncogenic HPV types with additional 2 probably/limited oncogenic types that are included in commonly used HPV testing platforms have widely varying impact numbers, both for the number needed to screen and the number needing follow-up.

The impact numbers, based on and in addition to the existing knowledging of HPV type-specific prevalence and oncogenicity, provide more integrated and explicit information in order to calculate (i) the number of tests needed for benefit; and (ii) the number of follow-up visits needed for benefit. Screening visits and gynecological follow-up visits demand resources and may involve adverse effects and it is therefore desirable to design screening programs with as high impact as possible.

Today, there are many HPV screening platforms that include extended HPV genotyping. If the NNS for a particular HPV type is high, meaning that the impact of the type to the screening program is low, the program may consider evaluating whether that type needs to be included

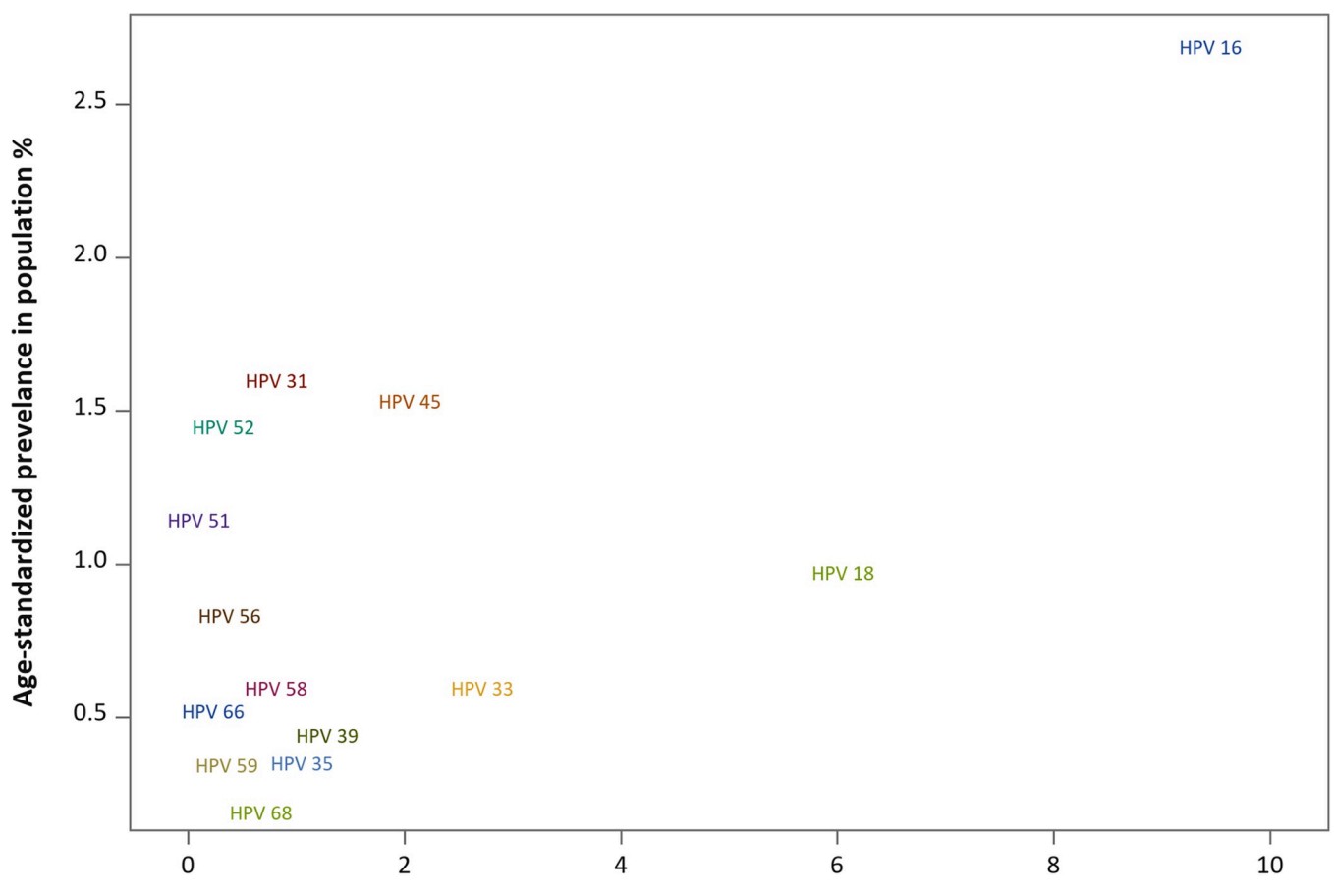

**Fig 4. Population prevalence of 14 HPV types and number of cases per 1,000 women positive for each type, in the pseudo-scenario that screening was absent.** HPV, human papillomavirus.

**Table 1. Age-specific HPV type distribution of invasive cervical cancer cases by screening status in last 10 years.**

| | Age <30 years | | Age 30–39 years | | Age 40–49 years | | Age 50–64 years | | Age > = 65 years | |
|---|---|---|---|---|---|---|---|---|---|---|
| | N | %[d] | N | %[d] | N | %[d] | N | %[d] | N | %[d] |
| Cases that were **screened** in the last 10 years | | | | | | | | | | |
| HPV 16 | 81 | 63.3 | 305 | 59.2 | 208 | 48.3 | 197 | 47.0 | 70 | 42.7 |
| HPV 18 | 34 | 26.6 | 121 | 23.5 | 98 | 22.7 | 73 | 17.4 | 17 | 10.4 |
| HPV 45 | 4 | 3.1 | 36 | 7.0 | 39 | 9.0 | 40 | 9.5 | 4 | 2.4 |
| Intermediate oncogenic types[a] | 8 | 6.3 | 30 | 5.8 | 29 | 6.7 | 29 | 7.0 | 24 | 14.6 |
| Lower oncogenic types[b] | 1 | 0.8 | 9 | 1.8 | 21 | 4.9 | 13 | 3.0 | 5 | 3.0 |
| Oncogenic HPV negative[c] | 0 | 0.0 | 14 | 2.7 | 36 | 8.4 | 67 | 16.0 | 44 | 26.8 |
| Total | 128 | 100 | 515 | 100 | 431 | 100 | 419 | 100 | 164 | 100 |
| Cases that were **unscreened** in the last 10 years | | | | | | | | | | |
| HPV 16 | 29 | 74.4 | 69 | 69.0 | 80 | 58.0 | 135 | 54.7 | 297 | 44.4 |
| HPV 18 | 6 | 15.4 | 15 | 15.0 | 26 | 18.8 | 29 | 11.7 | 51 | 7.6 |
| HPV 45 | 3 | 7.7 | 6 | 6.0 | 10 | 7.2 | 24 | 9.7 | 31 | 4.6 |
| Intermediate oncogenic types[a] | 1 | 2.6 | 6 | 6.0 | 9 | 6.5 | 26 | 10.4 | 105 | 15.6 |
| Lower oncogenic types[b] | 0 | 0.0 | 2 | 2.0 | 3 | 2.1 | 14 | 5.6 | 51 | 7.4 |
| Oncogenic HPV Negative[c] | 0 | 0.0 | 2 | 2.0 | 10 | 7.2 | 19 | 7.7 | 134 | 20.0 |

*(Continued)*

**Table 1.** (Continued)

|  | Age <30 years | | Age 30–39 years | | Age 40–49 years | | Age 50–64 years | | Age > = 65 years | |
|---|---|---|---|---|---|---|---|---|---|---|
|  | N | %[d] | N | %[d] | N | %[d] | N | %[d] | N | %[d] |
| Total | 39 | 100 | 100 | 100 | 138 | 100 | 247 | 100 | 669 | 100 |

[a]Intermediate oncogenic types include HPV 31, 33, 52, 58 (etiological fraction>2% according to IARC's data [4]).

[b]Lower oncogenic types include HPV 35, 39, 51, 56, 59, 66, 68 (etiological fraction <2% according to IARC's data [4]).

[c]Oncogenic HPV negative in formalin-fixed paraffin-embedded tumor blocks.

[d]Column percentage: number of cases of a certain type divided by all cases (oncogenic HPV negative cases included in the denominator).

HPV, human papillomavirus; IARC, International Agency for Research on Cancer.

in the screening. Similarly, the NNF data could be used to design referral strategies. Options could be, e.g., to directly refer women with HPV types with low NNF to gynecological examination, but require repeat testing or other triaging before referring women positive for HPV types with high NNF.

The impact numbers other than HPV16, especially the intermediate and lower oncogenicity types, tended to vary greatly by age. Among older women, these impact numbers were low with narrow CIs, whereas among younger women these impact numbers tended to be high

**Table 2. Screening preventable cervical cancer cases by HPV type.**

|  | Screened in last 10 years | | Unscreened in last 10 years | | | |
|---|---|---|---|---|---|---|
| Age-standardized incidence rate[a] | 8.4 | | 25.6 | | | |
| Number of cases | 3,609[b] | | 11,000[c] | | | |
| HPV type | Age-standardized % of cases[d] | Estimated number of cases[b] | Age-standardized % of cases[d] | Estimated number of cases[c] | Number of cases preventable[e] | % of cases preventable (95% CI)[f] |
| HPV 16 | 50.61 | 1,826 | 58.39 | 6,422 | 4,596 | 71.6 (69.1–73.9) |
| HPV 18 | 18.85 | 680 | 13.38 | 1,471 | 791 | 53.8 (40.6–63.1) |
| Other oncogenic types | 16.72 | 603 | 20.18 | 2,219 | 1,616 | 72.8 (66.8–77.4)[g] |
| Oncogenic HPV negative on tumor block | 13.82 | 498 | 8.06 | 886 | 388 | 43.8 (27.4–55.8) |

There were 4,254 invasive cervical cancer cases in Sweden during 2002–2011. Based on the age-standardized incidence rate in the population and among screened and unscreened women, we estimated that there would still have been 3,609 cervical cancer cases if all women had been screened and 11,000 cases if all women were unscreened. By using the age-standardized distribution of HPV16, 18, and other HPV types by screening history, we calculated the estimated number of cases by HPV type in pseudo-scenarios if all women were screened and if all women were unscreened. The difference between these 2 numbers represents the number of preventable cases.

[a]In women aged 20 years and above (per 100,000 person-years). Standardization was based on Swedish population in 2000.

[b]Estimated in pseudo-scenario that all women were screened.

[c]Estimated in pseudo-scenario that all women were unscreened.

[d]Number of cases of a certain type divided by all cases (oncogenic HPV–negative cases included in the denominator). Received from HPV genotyping of 2,850 out of 4,254 cases during 2002–2011. Age-standardized based on the Swedish population in 2000.

[e]Difference between the estimated number of cases in pseudo-scenario that all women were unscreened in last 10 years and the estimated number of cases in pseudo-scenario that all women were screened in last 10 years.

[f]Number of cases being prevented, divided by estimated number of cases in pseudo-scenario that all women were unscreened in last 10 years. CI were calculated through bootstrap sampling of 1,000 re-sampling with replacement.

[g]The results for HPV45, combined HPV31,33,52,58, and combined HPV35, 39, 51, 56, 59, 66, 68 are 72.8%, 70.1%, and 79.4%, respectively, and the CIs are largely overlapping. Due to small number of cases, they were combined into "Other oncogenic types."

CI, confidence interval; HPV, human papillomavirus.

**Table 3. Distribution of 14 high-risk HPV types among screened, unscreened cases and the population; impact numbers to prevent or detect one cervical cancer case by HPV type (CIs are presented in Fig 5 and S1A Table).**

| | % Among screened cases[a] | % Among unscreened cases[b] | % Among population[c] | Estimated number of cases screened[d] | Estimated number of cases unscreened[e] | Estimated number of women in population[f] | Population impact number (number needed to screen[k]) | | Number needed to follow-up (in test positive ones) | |
|---|---|---|---|---|---|---|---|---|---|---|
| | | | | | | | To prevent one case[g] | To detect one case[h] | To prevent one case[i] | To detect one case[j] |
| HPV 16 | 50.61 | 58.39 | 2.67 | 182.7 | 641.7 | 67,815 | 5,527 | 13,885 | 147 | 371 |
| HPV 18 | 18.85 | 13.38 | 0.96 | 68.1 | 147.0 | 24,275 | 32,125 | 37,280 | 307 | 356 |
| HPV 31 | 3.47 | 3.06 | 1.58 | 12.5 | 33.6 | 40,194 | 120,247 | 202,547 | 1,905 | 3,208 |
| HPV 33 | 2.87 | 3.69 | 0.58 | 10.3 | 40.6 | 14,711 | 83,884 | 245,200 | 486 | 1,421 |
| HPV 35 | 0.14 | 0.91 | 0.34 | 0.5 | 10.0 | 8,516 | 267,045 | 5,100,308 | 896 | 17,120 |
| HPV 39 | 0.70 | 1.28 | 0.43 | 2.5 | 14.0 | 10,852 | 220,334 | 1,008,712 | 942 | 4,314 |
| HPV 45 | 6.05 | 7.28 | 1.52 | 21.9 | 80.0 | 38,472 | 43,655 | 116,091 | 662 | 1,760 |
| HPV 51 | 0.13 | 0.03 | 1.13 | 0.5 | 0.3 | 28,630 | $\infty$[l] | 5,239,990 | $\infty$[l] | 59,133 |
| HPV 52 | 1.58 | 1.12 | 1.43 | 5.7 | 12.4 | 36,336 | 380,757 | 445,906 | 5,453 | 6,386 |
| HPV 56 | 0.81 | 0.75 | 0.82 | 2.9 | 8.2 | 20,705 | 479,556 | 871,466 | 3,913 | 7,112 |
| HPV 58 | 0.28 | 1.09 | 0.58 | 1.0 | 12.0 | 14,711 | 230,175 | 2,533,751 | 1,334 | 14,692 |
| HPV 59 | 0.56 | 0.36 | 0.33 | 2.0 | 3.9 | 8,312 | 1,339,680 | 1,252,334 | 4,389 | 4,103 |
| HPV 66 | 0.10 | 0.30 | 0.50 | 0.4 | 3.3 | 12,779 | 852,629 | 7,065,580 | 4,294 | 35,589 |
| HPV 68 | 0.04 | 0.31 | 0.18 | 0.1 | 3.4 | 4,453 | 783,010 | 18,162,183 | 1,374 | 31,878 |

[a]Age-standardized proportion of HPV types among cervical cancer cases that were screened in the last 10 years.

[b]Age-standardized proportion of HPV types among cervical cancer cases that were unscreened in the last 10 years.

[c]Age-standardized proportion of HPV types in women population.

[d]Estimated number of cases in 2011 in the pseudo-scenario that all women were screened. Calculated by number of cases per year during 2002–2011 (425) multiplied by incidence rate ratio between screened and the population (8.4/9.9 as shown in Fig 2) and multiplied by age-standardized proportion of HPV types among cervical cancer cases that were screened in the last 10 years.

[e]Estimated number of cases in 2011 in the pseudo-scenario that all women were unscreened. Calculated by number of cases per year during 2002–2011 (425) multiplied by incidence rate ratio between unscreened and the population (25.6/9.9 as shown in Fig 2), multiplied by age-standardized proportion of HPV types among cervical cancer cases that were unscreened in the last 10 years.

[f]Calculated as total number of women aged 23–64 years (2,536,995) multiplied by age-standardized proportion of HPV types in women population.

[g]Calculated as total number of women aged 23–64 years (2,536,995) divided by the difference between number of cases in the pseudo-scenario that all women were unscreened and number of cases in the pseudo-scenario that all women were screened

[h]Calculated as total number of women aged 23–64 years (2,536,995) divided by number of cases in the pseudo-scenario that all women were screened.

[i]Calculated as estimated number of women in population for each HPV type divided by the difference between number of cases in the pseudo-scenario that all women were unscreened and number of cases in the pseudo-scenario that all women were screened.

[j]Calculated as estimated number of women in population for each HPV type divided by number of cases in the pseudo-scenario that all women were screened.

[k]Interpretation should be that among every X number of women in screening population, one cervical cancer case caused by a certain HPV type can be detected or prevented.

[l]Infinity.

CI, confidence interval; HPV, human papillomavirus.

with very wide CIs. These wide CI were due to very few cases with those types among young women to be detected, and very few cases with those types in the upcoming 10 years to be prevented. The small number of cases itself suggests a low importance of these types in this age group. This suggests that strategies where only selected types are screened for or followed-up may be appropriate in younger age groups. In particular, consideration of no screening or no follow-up in women under 30 years of age may be warranted for the lower oncogenic types HPV35, 39, 51,56, 59, 66, and 68. Screening is always an ethical balance between the benefit and the adverse effects (unnecessary stress, unnecessary treatment linked to increased risk of

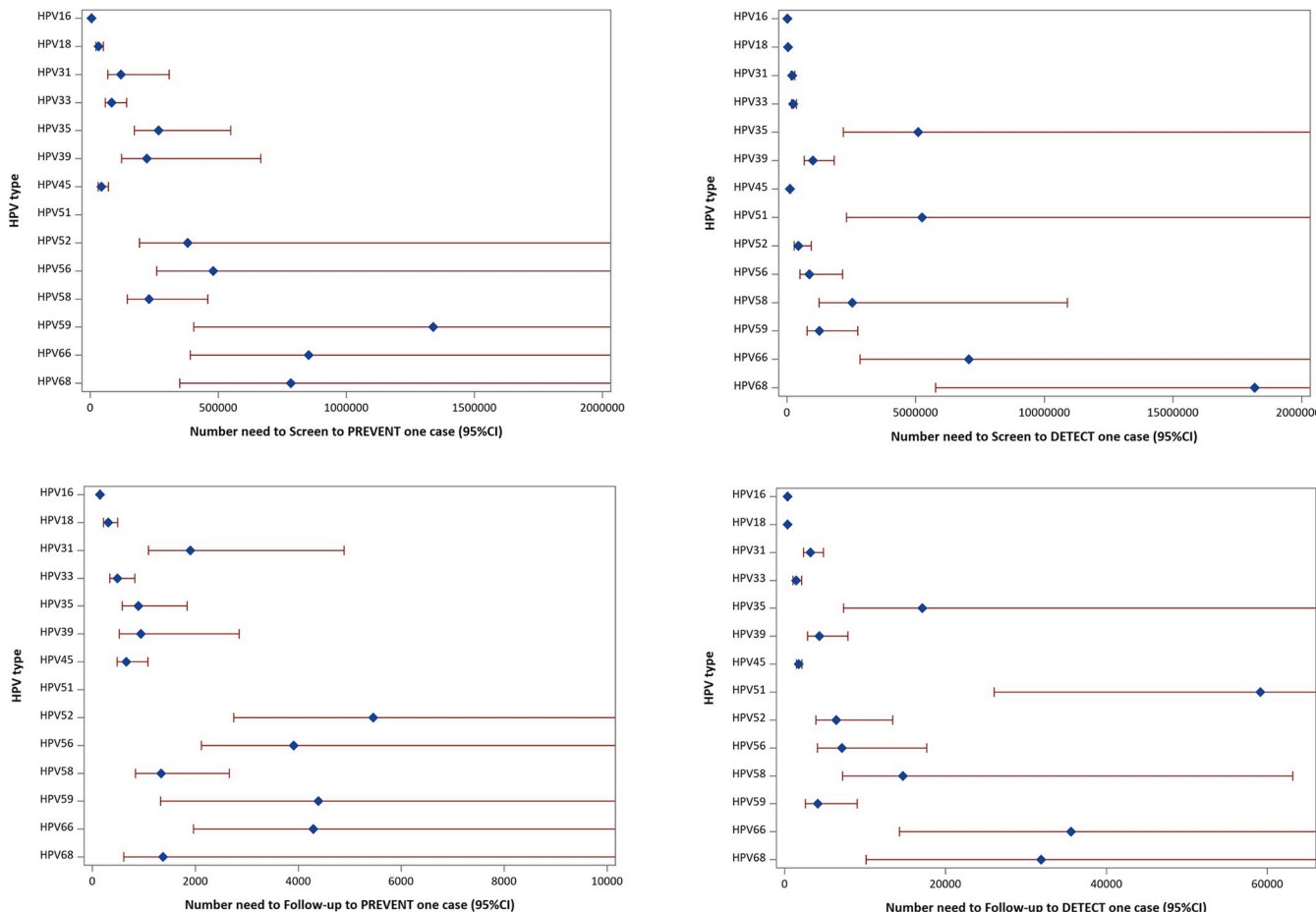

**Fig 5. Impact numbers (dots) and 95% CIs (lines) to prevent or detect one cervical cancer case by HPV type.** CI, confidence interval; HPV, human papillomavirus.

negative birth outcomes affecting particularly young women [28,29], etc.). The different impacts of screening for different HPV types have several ethical aspects. For example, if it is well motivated to screen for some types but not for others, screening for all of them as a package without informing the women about the different impacts is ethically questionable. Knowledge about the different impacts of different types could, e.g., be used when selecting an HPV test with an optimal impact. If an HPV test is being used that tests for HPV types with low oncogenicity, it would seem appropriate to inform the women about the associated risk of specific HPV types and whether the risk warrants a follow-up or not.

The strengths of the study is that it integrated longitudinal individual-level data of cervical screening and cervical cancer using large cohorts and the entire population of Sweden to estimate the HPV type-specific impact numbers. The impact number is a simplified indicator including the resource-benefit quantification integrating the factors of HPV type-specific prevalence, oncogenicity, and screening effectiveness. Certain factors are stable and some may vary across settings or over time.

The oncogenicity of different HPV genotypes is a biological property and thus does not differ between settings and over time. The varying oncogenicity of HPV types in relation to cervical cancer is well established from international studies on number of cervical cancer cases by HPV type, as well as comparing how common each type is in pre-cancer or cancer in relation

**Table 4. Impact numbers—number needed to screen and follow-up to prevent or detect one cervical cancer case by HPV type and age group at screening (CIs are presented in Fig 6 and S1B Table).**

| Age at screening | To prevent one case[a] | | | | To detect one case[b] | | | |
|---|---|---|---|---|---|---|---|---|
| | Age 23–30 years | Age 31–40 years | Age 41–50 years | Age 51–60 years | Age 23–30 years | Age 31–40 years | Age 41–50 years | Age 51–60 years |
| **Numbers needed to screen** | | | | | | | | |
| HPV 16 | 4,747 | 4,808 | 4,959 | 5,114 | 24,518 | 11,409 | 20,674 | 25,527 |
| HPV 18 | 50,908 | 14,811 | 23,522 | 22,367 | 49,036 | 29,589 | 45,342 | 71,477 |
| HPV 45 | 62,546 | 36,764 | 25,137 | 43,635 | 261,524 | 95,775 | 107,841 | 127,637 |
| Intermediate oncogenic types[c] | 52,212 | 53,493 | 30,049 | 15,460 | 261,524 | 125,498 | 128,714 | 170,182 |
| Lower oncogenic types[d] | 221,345 | 64,862 | 52,585 | 47,338 | 1,176,857 | 330,859 | 221,674 | 297,819 |
| **Numbers needing follow-up** | | | | | | | | |
| HPV 16 | 289 | 142 | 68 | 58 | 1,491 | 336 | 285 | 292 |
| HPV 18 | 1,204 | 137 | 123 | 87 | 1,160 | 274 | 237 | 278 |
| HPV 45 | 2,007 | 570 | 236 | 360 | 8,393 | 1,486 | 1,011 | 1,054 |
| Intermediate oncogenic types[c] | 4,586 | 2,297 | 780 | 353 | 22,972 | 5,388 | 3,343 | 3,887 |
| Lower oncogenic types[d] | 16,825 | 2,533 | 1,242 | 982 | 89,457 | 12,920 | 5,237 | 6,178 |

[a]Preventable cases in each age group are defined as the difference between estimated number of cases in the next age group in the pseudo-scenario that all women were unscreened and estimated number of cases in the next age group in the pseudo-scenario that all women were screened. For age group 51–60, the preventable cases are defined as all estimated of cases at ages 61–80, regardless of further screening history.

[b]Detected cases in each age group are defined as estimated number of cases in this age group in the pseudo-scenario that all women were screened.

[c]Intermediate oncogenic types include HPV 31, 33, 52, 58 (etiological fraction >2% according to IARC's data [4]).

[d]Lower oncogenic types include HPV 35, 39, 51, 56, 59, 66, 68 (etiological fraction <2% according to IARC's data [4]).

CI, confidence interval; HPV, human papillomavirus; IARC, International Agency for Research on Cancer.

to HPV positivity among women with normal cytology [5,30,31]. Well-established studies have compared HPV types and risk of histological diagnoses of Cervical Intraepithelial Neoplasia grade 2, grade 3, and worse (CIN2+ and CIN3+) [32–35]. The ranking of the 14 HPV types in our risk profile results under the no-screening scenario as well as NNF, which reflect oncogenicity, is to a large extent in line with the prior research findings.

The prevalence of different HPV genotypes may vary across settings and over time especially following the impact of HPV vaccination. Prevalence of a certain HPV type affects its NNS. International data has shown that HPV type-specific prevalences differ in different populations [6]. NNS can readily be recalculated in settings with different HPV type distributions. HPV vaccination is expected to largely supress the prevalence of HPV16, 18, 31, 33, 45, 52, and 58. The currently presented NNS is calculated in a population barely affected by vaccination, according to the year of HPV vaccine introduction, eligible age for vaccination and the age and calendar period for data retrieval. The NNS for HPV types that are vaccinated against are expected to increase with increased coverage of HPV vaccination in a population. HPV vaccinated birth cohorts are entering cervical screening, and the low efficiency of screening among young, vaccinated women has repeatedly been pointed out [36,37]. NNS will therefore need to be constantly monitered along with the population HPV vaccine coverage in each age group in order to obtain timely data on impact by HPV type.

Effectiveness of screening determines the case detection and prevention magnitude of NNS and NNF. Although it may vary across settings, the same recommendations for screening modality and intervals are recommended globally. As Sweden follows the global screening recommendations, our results on the screening prevention potential as well as the impact

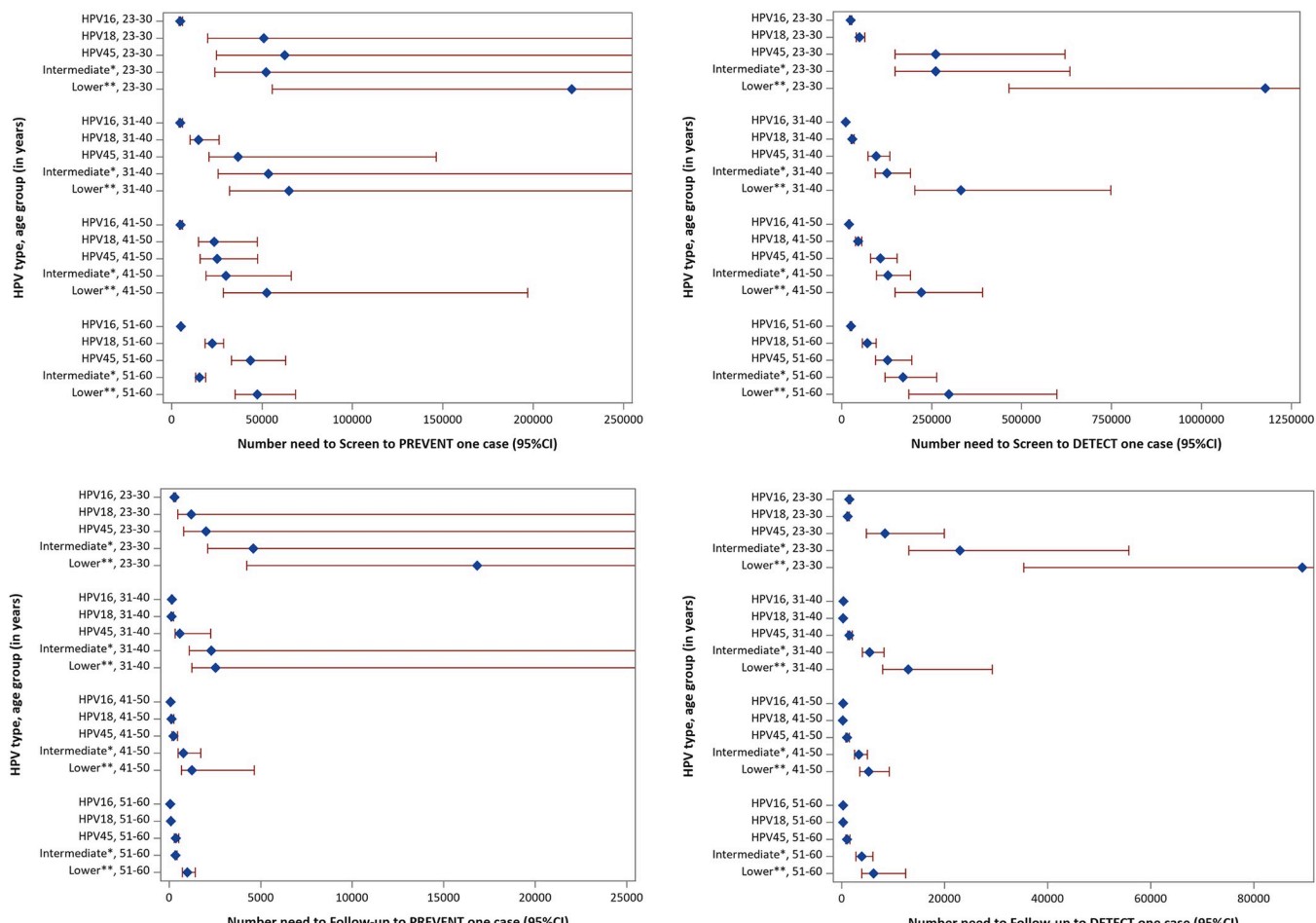

**Fig 6. Impact numbers (dots) and 95% CIs (lines) to prevent or detect one cervical cancer case by HPV type and age group.** * Intermediate oncogenic types include HPV 31, 33, 52, 58 [4]. **Lower oncogenic types include HPV 35, 39, 51, 56, 59, 66, 68 [4]. CI, confidence interval; HPV, human papillomavirus.

numbers for cancer prevention and detection should be valid in other settings that also follow the global screening recommendations and be useful also for countries considering adopting the global screening recommendations. It is worth mentioning that the screening effectiveness in this study was mainly evaluated in the era of cytology-based screening. Cytology remains the main triage test in HPV-based screening programs and the management and treatment of screen-detected cancer precursors is the same. Improving effectiveness of cervical screening is a constant effort. For example, our finding that historical screening prevented only 54% of HPV18-carrying cancers, implies that improved clinical management to enhance cancer prevention, particularly for adenocarcinoma (largely related to HPV18 [3,4]), should be pursued in particular for HPV18–positive women in HPV-based screening. Data from a European HPV-based screening trial already showed that HPV-based screening tended to have a greater gain in preventive effect for adenocarcinoma as compared to squamous-cell carcinoma (gain 69% (95% CI [31%, 86%]) versus 22% (95% CI [51%, −25%], respectively) [1]. In the future when there is substantial improvement of screening effectiveness, the impact numbers will need to be reexamined.

A major limitation of the study was using several assumptions to integrate data over different calendar periods and ages, which was described and discussed in the Methods section.

Another limitation in our calculation of impact number is, in younger age groups, that certain HPV types, particularly HPV51, had very few or even no cases, which hindered the estimation for NNS and NNF for cancer prevention. Nevertheless, since these types caused very few cases regardless of screening history, they are potentially negligible types to screen for. A further limitation is, regarding the cervical cancer cases being negative for oncogenic HPVs in tumor blocks, we know little about their HPV infection status and types in the years before cervical cancer diagnosis, thus we cannot predict how HPV-based screening would impact this group. We will be able to know more in the near future with accumulated data from HPV-based screening. Nonetheless, since few cervical cancer cases in young women were negative for oncogenic HPVs in tumor blocks, our estimates for young women were barely affected by this issue.

To conclude, the 12 oncogenic HPV types and 2 commonly tested probably oncogenic types have large variations in their impact numbers, and thus different cervical cancer screening efficiency. Use of HPV screening tests that focus on HPV types with low number needed to screen and referral algorithms focusing on HPV types with low number needing follow-up may be considered, especially for younger women. To increase screening effectiveness, strategies to follow-up HPV18 positivity may need to be improved. Impact numbers can be monitored over time, following the change of HPV type-specific prevalence and screening effectiveness, to timely provide data to the screening program for consideration of possible adjustment.

## Supporting information

**S1 STROBE Checklist. STROBE checklist.**
(DOCX)

**S1 Appendix. Organized cervical screening program and registry in Sweden.**
(DOCX)

**S1 Table. Impact numbers and bootstrap confidence intervals (CI) for detecting and preventing one cervical cancer case by human papillomavirus (HPV) type.**
(DOCX)

**S1 Protocol. Protocol for cervical cancer age-specific incidence by screening history and HPV genotyping of cervical cancer cases, as protocols in ethical application.**
(PDF)

**S2 Protocol. Randomized implementation of primary human papillomavirus (HPV) testing in the organized screening for cervical cancer in Stockholm.**
(PDF)

## Author Contributions

**Conceptualization:** Jiangrong Wang, K. Miriam Elfström, Karin Sundström, Joakim Dillner.

**Data curation:** Jiangrong Wang, K. Miriam Elfström, Camilla Lagheden, Carina Eklund, Pär Sparén.

**Formal analysis:** Jiangrong Wang.

**Funding acquisition:** Joakim Dillner.

**Investigation:** Jiangrong Wang, Joakim Dillner.

**Methodology:** Jiangrong Wang, Joakim Dillner.

**Resources:** Camilla Lagheden, Carina Eklund, Joakim Dillner.

**Supervision:** Karin Sundström, Pär Sparén, Joakim Dillner.

**Validation:** K. Miriam Elfström.

**Visualization:** Jiangrong Wang.

**Writing – original draft:** Jiangrong Wang.

**Writing – review & editing:** Jiangrong Wang, K. Miriam Elfström, Camilla Lagheden, Carina Eklund, Karin Sundström, Pär Sparén, Joakim Dillner.

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
