## [Editor Report · Decision Letter 0]

6 Apr 2023

Dear Dr Dillner, 

Thank you for submitting your manuscript entitled "HUMAN PAPILLOMAVIRUS TYPE-SPECIFIC IMPACT NUMBERS IN CERVICAL SCREENING" for consideration by PLOS Medicine.

Your manuscript has now been evaluated by the PLOS Medicine editorial staff and I am writing to let you know that we would like to send your submission out for external peer review.

Please re-submit your manuscript within two working days, i.e. by Apr 10 2023 11:59PM.

Kind regards,

Philippa Dodd, MBBS MRCP PhD

PLOS Medicine

---

## [Decision Letter · Decision Letter 1]

25 May 2023

Dear Dr. Dillner,

Thank you very much for submitting your manuscript "HUMAN PAPILLOMAVIRUS TYPE-SPECIFIC IMPACT NUMBERS IN CERVICAL SCREENING" (PMEDICINE-D-23-00933R1) for consideration at PLOS Medicine. 

Your paper was evaluated by an associate editor and discussed among all the editors here. It was also discussed with an academic editor with relevant expertise, and sent to independent reviewers, including a statistical reviewer. The reviews are appended at the bottom of this email and any accompanying reviewer attachments can be seen via the link below:

[LINK]

In light of these reviews, I am afraid that we will not be able to accept the manuscript for publication in the journal in its current form, but we would like to consider a revised version that addresses the reviewers' and editors' comments. Obviously we cannot make any decision about publication until we have seen the revised manuscript and your response, and we plan to seek re-review by one or more of the reviewers. 

We expect to receive your revised manuscript by Jun 15 2023 11:59PM. Please email us (plosmedicine@plos.org) if you have any questions or concerns.

We look forward to receiving your revised manuscript. 

Sincerely,

Alexandra Schaefer, PhD

PLOS Medicine

plosmedicine.org

ACADEMIC EDITOR COMMENTS

It's not that well-written in my opinion. In particular, I found it very hard to follow the methods - it is really hard to know when actual empirical data are being used and presented and when they are based on assumptions. I would like to see this much more clearly in the revised version. I've made some specific comments below.

Title: ‘impact numbers’ not meaningful – this term should be revised in the title and throughout the manuscript.

Abstract: will probably need to be rewritten, based on reviewer 4’s review – but as it stands, it’s hard to understand

Materials and methods, p4: This section starts ‘This study was based on the entire population

of women live [sic] in Sweden from 1990s to 2019…’ But some data sources are for sub-populations and estimates are presumably extrapolated. Could the authors a) make it clearer from the beginning that data sources are not all for the whole population and b) say what the age range is that defines ‘all women’?

Materials and methods, p5: ‘by screening history’ – this seems to be applying an assumption, rather than being based on women’s actual screening history. Is this correct? If so, can it be more explicit? On p4, it says that each woman’s screening history was retrieved – so how were these data used? It would actually be very helpful to have a table that lists the actual data sources and says how these data are then used.

Tables, figures: Age groups should be reported consistently. E.g. Fig 1, first box, Swedish women aged 23-84; left had side box has 25-84; middle boxes – there are numbers stratified by different age groups – please explain in legend. Table 1 start at <30 – is this from age 23 or 25? And >=65, is this up to 84 or all over that age? Table 4, why are preventable cases 61-80 included in age group 51-60? If oldest age group is 84 elsewhere, why not here?

Discussion: use of the term ‘impact numbers’ needs to be revised because it’s confusing. E.g. para 2, lines 3-4, ‘A program could e.g. decide to use a screening test that does not test for HPV types with high impact numbers’ I think it means if the numbers NNS and NNF are high – but it sounds like the opposite, that the HPV type wouldn’t be included if the impact of testing for it were high.

Discussion, p8, para 3: ‘This suggests that strategies where only selected types are screened for or followed-up may be particularly appropriate in younger age groups.’ It would be good to see a bit about the ethical aspects and public communication of deciding not to screen for an oncogenic HPV type that is included in a test and for which a result would be available, even if the number needed to screen is high.

GENERAL COMMENTS

Please respond to all editor and reviewer comments.

Please include page numbers and line numbers in the manuscript file. Use continuous line numbers (do not restart the numbering on each page).

Please cite the reference numbers in square brackets (e.g., “We used the techniques developed by our colleagues [19] to analyze the data”). Citations should be preceding punctuation.

Please cite your Supporting Information as outlined here: https://journals.plos.org/plosmedicine/s/supporting-information

Please ensure consistency in your number format and revise your manuscript accordingly including the Supplementary Material (51·4% or 51.4%; 1000 or 1,000).

Please remove the Statements section from your main manuscript. The data should only be included in the corresponding section in the online submission form. 

TITLE

Please revise your title according to PLOS Medicine's style. Your title must be nondeclarative and not a question. It should begin with main concept if possible. "Effect of" should be used only if causality can be inferred, i.e., from an RCT. Please place the study design in the subtitle (ie, after a colon).

ABSTRACT

Please structure your abstract using the PLOS Medicine headings (Background, Methods and Findings, Conclusions). The Methods and Findings sections should be combined into one section, “Methods and findings”."

Please define HPV at first in the abstract.

Abstract Background:

*Provide the context of why the study is important. The final sentence should clearly state the study question.

Please ensure that all numbers presented in the abstract are present and identical to numbers presented in the main manuscript text.

PLOS Medicine requests that main results are quantified with 95% CIs as well as p values. Please include. When reporting p values please report as p<0.001 and where higher as the exact p value p=0.002, for example. For the purposes of transparent data reporting, if not including the aforementioned please clearly state the reasons why not.

Please include any important dependent variables that are adjusted for in the analyses.

Throughout, suggest reporting statistical information as follows to improve clarity for the reader “22% (95% CI [13%,28%]; p</=)”. Please amend throughout the abstract and main manuscript.

Please note the use of commas to separate upper and lower bounds, as opposed to hyphens as these can be confused with reporting of negative values.

Please define all abbreviations used for statistical reporting at first use.

When a p value is given, please specify the statistical test used to determine it.

Please provide brief demographic details of the study population (e.g. sex, age, ethnicity, etc)

In the last sentence of the Abstract Methods and Findings section, please describe the main limitation(s) of the study's methodology.

Abstract Conclusions:

*Please begin your Abstract Conclusions with "In this study, we observed ..." or similar, to summarize the main findings from your study, without overstating your conclusions. Please emphasize what is new and address the implications of your study, being careful to avoid assertions of primacy. 

*Please avoid vague statements such as ""these results have major implications for policy/clinical care"". Mention only specific implications substantiated by the results.

AUTHOR SUMMARY

At this stage, we ask that you include a short, non-technical Author Summary of your research to make findings accessible to a wide audience that includes both scientists and non-scientists. The Author Summary should immediately follow the Abstract in your revised manuscript. This text is subject to editorial change and should be distinct from the scientific abstract. Please see our author guidelines for more information: https://journals.plos.org/plosmedicine/s/revising-your-manuscript#loc-author-summary.

The summary should include 2-3 single sentence, individual bullet points under each of the questions.

It may be helpful to review currently published articles for examples which can be found on our website here https://journals.plos.org/plosmedicine/

INTRODUCTION

p.3: Please define WHO in the Introduction.

p.3: Please define HPV in the Introduction.

p.3: Please define IARC in the Introduction.

Please address past research and explain the need for and potential importance of your study. Indicate whether your study is novel and how you determined that. If there has been a systematic review of the evidence related to your study (or you have conducted one), please refer to and reference that review and indicate whether it supports the need for your study. 

Please remove the study results and/or conclusion from the Introduction.

Please conclude the Introduction with a clear description of the study question or hypothesis.

METHODS AND RESULTS

Please ensure that the study is reported according to the STROBE guideline, and include the completed STROBE checklist as Supporting Information. Please add the following statement, or similar, to the Methods: "This study is reported as per the Strengthening the Reporting of Observational Studies in Epidemiology (STROBE) guideline (S1 Checklist)."

Did your study have a prospective protocol or analysis plan? Please state this (either way) early in the Methods section.

For all observational studies, in the manuscript text, please indicate: (1) the specific hypotheses you intended to test, (2) the analytical methods by which you planned to test them, (3) the analyses you actually performed, and (4) when reported analyses differ from those that were planned, transparent explanations for differences that affect the reliability of the study's results. If a reported analysis was performed based on an interesting but unanticipated pattern in the data, please be clear that the analysis was data-driven.

PLOS Medicine requests that main results are quantified with 95% CIs as well as p values. Please include. When reporting p values please report as p<0.001 and where higher as the exact p value p=0.002, for example. For the purposes of transparent data reporting, if not including the aforementioned please clearly state the reasons why not.

Please include any important dependent variables that are adjusted for in the analyses.

Suggest reporting statistical information as detailed above – see under ABSTRACT

p.4: Please change “This study was based on entire population of women live in Sweden from 1990s to 2019, […]” to “This study was based on the entire population of women living in Sweden from the 1990s to 2019, […]”. Please check your manuscript carefully for grammar, spelling and punctuation.

p.4: “We integrated these data sources through cross-linkage at the individual level using personal identification number, and generate fundamental parameters of […]” – please change “generate” to “generated” and ensure to keep the tense consistent throughout the manuscript.

p.4: Please define PCR in the Methods.

p.4: Please write “Table 1”/“Figure 1” instead of “table 1”/”figure 1” and check throughout your manuscript.

p.7: Please change “pseudo scenario” to “pseudo-scenario” and check carefully throughout your manuscript to keep a consistent spelling.

p.7: Please remove the comma in “Figure 4 compares the prevalence and risk profile across 14 HPV types in the pseudo scenario of no screening, in a two-dimensional graph”.

p.7: Please change “Y-axis displays the age-standardized prevalence in percentage of each type in the population, and x-axis displays the incidence of cervical cancer among women positive for each HPV type in absence of screening.” to “The y-axis displays the age-standardized prevalence in percentage of each type in the population, and the x-axis displays the incidence of cervical cancer among women positive for each HPV type in absence of screening.”.

p.7: Please re-write the sentence starting with “Screening likely to have contributed to around […]”. Editorial suggestion: Screening likely contributed to approximately 72% (69%-74%) reduction in cases caused by HPV16 and by types other than 16 and 18, whereas it contributed to only 54% (41%-63%) reduction in cases caused by HPV18 (Table 2).

p.7: “The impact numbers for cervical screening in the population, i.e NNS and NNF to detect or

prevent one case of cervical cancer, […]” – please change “i.e” to “i.e.”.

p.7: Please check the grammar and punctuation for the following sentence: “For HPV51 NNS and NNF could not be estimated because too few HPV51-positive cervical cancer cases were detected during 10 years in the country of Sweden) (Table 3, Figure 5, Supplementary Table 1).”

Please present numerators and denominators for percentages, at least in the Tables [not necessarily each time they're mentioned].

DISCUSSION

Please present and organize the Discussion as follows: a short, clear summary of the article's findings; what the study adds to existing research and where and why the results may differ from previous research; strengths and limitations of the study; implications and next steps for research, clinical practice, and/or public policy; one-paragraph conclusion.

Please remove all subheadings within your Discussion e.g., limitations and other considerations.

p.8: “In particular, the lower oncogenic types HPV35, 39, 51, 56, 59, 66, 68 may be considered not screen for or not directly refer to follow-up in women below age 30.” – please change to “In particular, the lower oncogenic types HPV35, 39, 51,56, 59, 66, 68 may not be considered for screening or direct follow-up in women under 30 years of age.”.

p.8: Please define CIN2+ and CIN3+ in the Discussion.

p.9: The third paragraph (In general, the presented impact numbers should largely be […]) seems redundant as it repeats the statements of the previous paragraphs. Please be careful to make key statements and avoid repetition.

p.9: “Nevertheless, as these types caused very few cases regardless of screening historyimply that they are potentially negligible types to screen for.” – please change to “Nevertheless, since these types caused very few cases, regardless of screening history, they imply that they are potentially negligible types to screen for.”.

FIGURES

For all Figures, please ensure that you have complied with our figures requirements http://journals.plos.org/plosmedicine/s/figures.

Please provide titles and legends for all figures (including those in Supporting Information files).

Please define abbreviations used in the table/figure legend of each figure and/or table (including those in Supporting Information files).

In Figure 2, please provide 95% CIs.

Figure 3: Please define the scale on the y-axis as “%” is not sufficient in this case.

Figure 4: The description of the different scales used in Figure 4 (left graph x-axis on regular scale, right graph x-axis on log scale) should not be in brackets in the title, but part of the figure description.

Please consider avoiding the use of red and green in order to make your figure more accessible to those with colour blindness.

Please indicate in the figure caption the meaning of the whiskers in Figure 5 and 6.

REFERENCES

Please ensure that journal name abbreviations match those found in the National Center for Biotechnology Information (NCBI) databases (http://www.ncbi.nlm.nih.gov/nlmcatalog/journals), and are appropriately formatted and capitalised.

Please also see https://journals.plos.org/plosmedicine/s/submission-guidelines#loc-references for further details on reference formatting. 

Where website addresses are cited, please specify the date of access. 

Comments from the reviewers:

Reviewer #1: The manuscript contains the calculations for relevant impact numbers of cervical cancer screening, referred to the Swedish population. Data on HPV type-specific prevalence (in cancer cases and in population), oncogenicity and screening effectiveness were collected from different groups and over more than 20 years. By incorporating these data in the analyses, the authors have calculated the number need to screen (NNS) and the number need to follow-up (NNF) to detect or to prevent a case of cervical cancer by each HPV type. These impact numbers were substantially different among the 14 HPV types evaluated (12 high-risk and 2 probably/possibly oncogenic). On the basis of the results, the authors provide a stratification of these HPV types (as high, intermediate and lower oncogenicity), and indicate (also by age-groups) the most effective screening strategy, that can imply no screening or no follow up for infections by the lower oncogenic types, especially among the youngest age groups. The approach allows recalculation of the impact numbers over time, and this is an important aspect. The cervical cancer screening is being implemented in many countries with high-riskHPV testing as primary test; moreover, the use of a risk-based approach to manage the women with a positive screening result (where HPV genotyping could play a role as triage test) and the integration of primary (anti-HPV vaccination) and secondary (screening) preventive measures need to implement the protocols and monitor the effectiveness of the screening. The authors claim that these results could apply to other countries with similar type-specific HPV prevalences and similar screening efficiency; generalizability will probably be limited by differences in the prevalence of different HPV types (observed also within the same country) and by the substantial differences in anti-HPV vaccination coverages. Anyhow, the topic of this paper is of great importance in the field of cervical cancer prevention, where instruments to device the best strategies to adapt and personalize the screening strategies is highly needed. Moreover, it is important to the health community at large because these changes to be effective need that all the healthcare professionals give updated and concordant informations to the women. 

MINOR OBSERVATION:

-TABLE 1: a footnote reporting the distribution of the HPV types in the three subgroups could be usefully added

Reviewer #2: The paper represtnts a clever use of existing data. It will change our way to think about genotyping in cervical cancer screening.

I have only few comments to improve the roprting and a comment on the interpretation that could be a matter for discussioon. 

Abstract

I do not understand why NNS and NNF are reported only for women below 30, while for the all ages is reported only the variation without any absolute number. For the reader is not easy to follow the results as reported in the abstract. 

I think it should be clear that protection estimates come from screening with Pap test, not HPV test. Instead of the generic disclaim "Although reasonable limits for impact numbers may differ between settings" a more specific comment on the different level of protection given by HPV now could be appropriate. 

Introduction

Clear and well written. I only suggest better explaining how the number need to detect and number need to prevent interacts: the two desirable effect of screening, i.e. early detection of cancers and prevention of cancers, are additive and there is no way that any positive event is counted twice. 

I have a comment about the type specific NNS. It can be high for two reasons, low prevalence in the population or low oncogenic risk. If the issue is low prevalence, the type specific NNS to screening is not relevant for the decision making of including it or not, since the real NNS is that for all types. The NNS of the overall screening test is not the average of the NNS of single HPV types. As a paradox if we use a taxonomy that splits the HPV 16 in different variants, also the NNS of each HPV 16 variant will increase dramatically, without changing nothing about the efficiency or effectiveness of the test. 

Methods

According to my understanding of the computations made by the authors, the underlying model is simple but robust; its simplicity is a strength because it is also transparent. Nevertheless, simplicity of the model implies many assumptions. I suggest to add a paragraph or a box reporting all the implicit assumptions made by your model. The main issues are two, from first glimpse of the described methods: 

1) the model assumes a steady state, i.e. the analysis of HPV types in screened and unscreened population assumes that these proportion are the product of a stable dynamic between infections and progression to cancers in the two populations (this assumption is acceptable but should be explicit); probably you are also assuming that incubation time is equal for all types. The fact that this assumption is not true is probably the explanation for a different ratio between HPV16 and HPV low oncogenic NNS and NNF across ages: the incubation time is different, so there are few cancers from low-oncogenic types in younger women.

2) how do you treat the large part of cancers that are not positive for any HPV oncogenic type? According to table 1 and 2, it seems that they are distributed in the proportions of 16, 18 and other oncogenic but maybe I am wrong. It would be better to make an explicit statement on this point. 

I do not understand why all the measures are "age standardized". Actually all the relevant parameters are age-specific and all the counterfactual scenarios are estimated on a "standard" population, I do not see the need for standardizing. Particularly for type-specific HPV prevalence, standardization may be relevant only for the comparison of the screened and non-screened population that have different age structures. If this is the rationale, I suggest to explain it. All the other parameters are always age specific.

Results

First line of the results and figure 2 reports incidence in 2004-2011, while table two reports cancers from 2002 to 2011. Can you clarify?

Please define the time of incidence is per 100,000/y or cumulative for the 8-year period? 

Table 2. Could you report also the protection of screening for HPV-negative cancers? 

Figure 2. y-axis report if the incidence is annual or cumulative.

Discussion 

pag 8 second paragraph, I do not agree with the sentence: "An alternative strategy to use the NNS data could e.g. be to decide that detection of HPV types with high impact numbers is a normal screening result". As discussed before, if we adopt a different taxonomy and we give a different name to all HPV16 variants, the NNS will increase for each of these, the same for HPV 45 in some populations where it is extremely rare, but it will be always worth to include it among the oncogenic ones. Furthermore, low prevalence of a HPV type increases the type specific NNS but not the burden of assessment tests. NNS is relevant to decide if we have to screen or not at all, but it is not relevant for deciding which type should be included in the set of HPV types to test. 

pag 9, second paragraph: when mentioning the limitation that the screening effectiveness assessment was conducted for cytology-based screening, I think it is worth to report how much is expected to improve the protection with HPV based test, particularly for adenocarcinoma; an estimate could be given based on the pooled analysis of European trials (reference n 1) .

Reviewer #3: Alex McConnachie, Statistical Review

This review considers the statistical elements of the paper by Wang et al, looking at impact numbers in cervical screening in Sweden, in relation to HPV type and age.

The paper is a very nice analysis of routinely collected data, and data collected as part of other studies, and an excellent demonstration of how this can be used to inform public health policy. The impact numbers and how they are calculated are explained in great detail. The use of bootstrapping to generate confidence intervals is appropriate. The presentation is, on the whole, very clear.

I have one or two comments, which are minor, bordering on the trivial.

The date range for the study population is a little vague; should exact dates be given?

The paragraph giving descriptions of the NNS and NNF to detect or prevent one case is very good, in that it accurately describes each parameter. However, it is quite dense. Could some consideration be given to presenting these as formulas? Would some readers find that easier to follow?

In Table 2, do the columns headed "Population" add anything? Could it be made clearer that the columns headed "Screened in last 10 years" actually relate to the hypothetical situation where everyone is screened (and similarly for the "Unscreened in last 10 years")? It is clear from the footnotes, but could perhaps be clearer in the headers.

In Table 4, the columns are labelled as "Number needed to screen to detect (or to prevent) one case". However, only the top half of the table reports NNS, and the bottom half gives NNF. Maybe the columns should simply say "To detect one case" and "To prevent one case".

In Figure 2, the abbreviation "ASR" is not defined.

In Figures 5 and 6, the confidence intervals are asymmetrical, and those to the left of each panel are quite compressed. Would the figures work better with the x-axes on a log scale? Each figure has a different range on the x-axis - would showing on a log scale allow these to have a common scale? Would that be slightly better?

Reviewer #4: The current revision of the paper includes an interpretation in the abstract that is non-committal. Focusing on the NSN and NNF for women over 30 years might be more useful for an interpretation to put money toward screening and following up the abnormals. 

Methods - it is inappropriate to classify HPV types by whether or not they are in the Merck based nonovalent vaccine- do not let industry drive your definitions, let the science. Why don't you group the HPV types in the way that BD has grouped them according to aggressiveness of oncogenesis with 16/18/31 as separate highest risk types, and then 33/58 together, then 52 and 45 separately as medium high risk, then type 51 alone as a lower risk HR type, then 35/39/68 grouped together, and 56/59/55 grouped together. At least this grouping of HPV types is associated with risk of CIN 3+ over time. 

What is your assumption about HPV vaccination? You are using the Swedish population and creating pseudo scenarios that are dependent on the current HPV vaccination type and uptake remaining constant. Is this true? How does this limit your conclusions?

Your definition of NNS and NNT are too simplistic. You are creating an ideal situation where you only value screening for follow up if a cancer is detected. Screening is always a BALANCE between over-screening and leaving cancers on the table. You might check with a Bayseian mathetmatician about the pseudo scenarios you have created that the level of error inherent in each.

Figur2-What percentage of the population did not have a screen prior to her diagnosis? Why was she diagnosed with cervical cancer? symptoms? incidental for something else? what is the age distribution of these cancers? 

Figure 4 is not offering any information other than a plot. 

Age stratification needs to be above 30, not above 50 - or you need to provide adequate justification for not using 30 and older for your analysis. 

Discussion section needs to state that the limitation of racial/ethnicity diversity in Sweden means these data cannot be extrapolated to other continents where the population homogeneity is dissimilar. 

HPV 18 discussion is interesting - was the screening cytology based? if we went to HPV primary testing would we have missed those cancers? if we went with HPV 18 testing followed by Dual Stain, would those cancers have been missed in screening? Is it a fault of the screening that these cancers were not detected, or was it a fault of the women not coming in for screening?

[LINK]

---

## [Decision Letter · Decision Letter 2]

4 Sep 2023

Dear Dr. Dillner,

Thank you very much for re-submitting your manuscript "Impact of different types of Human Papillomavirus in cervical screening: Population-based estimations" (PMEDICINE-D-23-00933R2) for review by PLOS Medicine.

I have discussed the paper with my colleagues and the academic editor and it was also seen again by two reviewers. I am pleased to say that provided the remaining editorial and production issues are dealt with we are planning to accept the paper for publication in the journal.

[LINK]

We look forward to receiving the revised manuscript by Sep 11 2023 11:59PM.   

Sincerely,

Alexandra Schaefer, PhD

Asscoiate Editor 

PLOS Medicine

plosmedicine.org

Requests from Editors:

GENERAL

Thank you for considered and detailed responses to editor and reviewer comments.

Please see below for further minor points that we request you respond to in full.

Please add 'years’ to ages stated throughout your manuscript.

We note that the Supporting Information file S2 (Figure S2) provides important details for the understanding of the paper. We leave it to your discretion whether you wish to include it in the main paper.

Thank you for providing a copy of your study protocol (S7) as a supporting information file. Please send this in the original language and an English translation. Please detail any deviations from this study protocol in the Methods section of your manuscript.

ACADEMIC EDITOR COMMENTS

Thank you for the opportunity to read the revised manuscript. The authors have made extensive changes and the description is much clearer. I apologise for the rather impolite comment about the writing in the previous version. The authors had addressed my main comments.

1. Introduction: line 83, “Impact numbers [12] are a suitable measurement.” Whilst these are described in the next para, it would help to at least introduce ‘population’ and ‘disease’ impact numbers because they are not widely used terms.

2. Line 144: spelling, ”assumPtions”

3. Line 155: wording, suggest “We found two publications suggesting…”

4. Line 177: “HPV vaccination has yet noticeably affected the study population…” should it be “…has NOT yet…”?

5. Fig 1: I cannot see where footnote c refers to

6. Lines 370-1: “NNS will therefore be constantly monitored…to timely adjust screening strategies.” This implies that a decision to choose which HPV types should be followed up has already been made. However, in response to Reviewer #2’s comment about HPV type-specific prevalence, the authors say, “Which conclusion to make is up to the program – we are simply providing data that can be considered.” Can the authors clarify whether this strategy has already been adopted?

7. Line 411: wording “…to timely inform the screening program for adjustment.” As above, this sentence needs to be clarified – and made grammatically correct.

EDITORIAL COMMENTS

When revising your manuscript, please consider that your manuscript needs to be accessible to a wide audience and aim to improve your writing for this purpose.

COMPETING INTEREST

You indicated that authors KS has received research grants from Merck and Co, LLC. For authors with ties to industry, please indicate whether any of the interests has a financial stake in the results of the current study.

DATA AVAILABILITY STATEMENT

Thank you for agreeing to make your data available. At this time, please provide the link to the data repository and accession numbers required for access.

TITLE

Since the title must be nondeclarative, we suggest changing your title to “Outcomes of cervical screening by human papillomavirus genotype: a population-based study in Sweden” or similar. We suggest changing the short title to “HPV type-specific outcomes in cervical screening” or similar.

ABSTRACT

l.18 suggest: “Cervical screening programs use testing for human papillomavirus (HPV) genotypes.”

l.32: Please change ‘can’ to ‘could’.

In the last sentence of the Abstract Methods and Findings section, please describe the main

limitation(s) of the study's methodology. The statement “However, it can readily be recalculated in other settings and monitored when HPV-type specific prevalences change.” should be included in the discussion (We suggest changing the statement to ““However, it can readily be recalculated in other settings and monitored when HPV-type specific prevalence changes.”).

Please revise your Abstract Conclusion. The statement that you found different screening impacts for different HPV types is very vague. Also, findings observed in your study should be presented in past tense. We suggest, “In this study, we observed that the impact of cervical cancer screening varies depending on the HPV type screened for. Estimating and monitoring the impact of screening by HPV type can facilitate the design of effective and efficient HPV-based cervical screening programs.” or similar.

AUTHOR SUMMARY

Your author summary requires revision. The summary should include 2-3 single sentence, individual bullet points under each of the questions (Why Was This Study Done? What Did the Researchers Do and Find? What Do These Findings Mean?). Please note that the text should be distinct from the scientific abstract. In the final bullet point of ‘What Do These Findings Mean?’, please describe the main limitations of the study in non-technical language.

It may be helpful to review currently published articles for examples which can be found on our website here https://journals.plos.org/plosmedicine/

INTRODUCTION

l.62: Please define ‘HPV’ at first use (you first define ‘HPV’ in l.64).

l.77: Please change ‘suggest’ to ‘suggests’.

l.102: Please exchange ‘hope’ with ‘aim’.

METHODS AND RESULTS

Please ensure that, when possible, the 95% confidence intervals quantified for the main results are reported in the main text. 

For description of age, please add ‘years’ (e.g. l.133 ‘women aged 23-29’). Please revise throughout the entire manuscript.

I.121 onwards: would benefit from being split into shorter paragraphs to improve reader accessibility.

l.122: Please change to “women in the capital region of Stockholm”.

l.144: Please change ‘assumtions’ to ‘assumptions’.

l.155: Please change to “We found two publications suggesting […]”.

l.163: Please change the format of ‘4254’ to ‘4,254’. Please ensure consistency in number format and revise throughout the entire manuscript (e.g. l.166 ‘2850’ or l.206 ‘1000’).

l.177-179: Please clarify whether HPV vaccination has or has not yet affected the study population. Editorial suggestion: “HPV vaccination has not yet had a noticeable impact on the study population included in this study: data were not available from birth cohorts of women vaccinated in the school-based or similar high-coverage HPV vaccination program.”

l.184: We note that you usually write the word “other” in quotation marks (here: other 12 types). Please revise and ensure to use a consistent format throughout the entire manuscript.

l.211-212: Please note that the citation does not meet the requirement of PLOS reference formatting. Please remove ‘Heller et. al'.

I.252: This statement should be presented early in the method section. Please move to perhaps follow the subheading at line 105.

I.274: It might be helpful to re-define the population size such that ‘30%’ can be quantified.

l.279: Please change ‘S2 Figure’ to ‘Figure S2’.

l.286: Please add a comma following ‘and 33’. 

ll.283-287: Please note that results should be described in past tense. 

I.302/305: Suggest ‘lower’ instead of ‘reduced’.

DISCUSSION

l.337: Please change ‘has’ to ‘have’.

l.341: Please change ‘does test’ to ‘tests’.

l.386: Please change ‘Data from European HPV-based screening trial’ to ‘Data from a European HPV-based screening trial’.

l.391: Please change ‘limitation of study’ to ‘limitation of the study’.

l.392: Please change ‘method’ to ‘methods’.

l.404: ‘Possibly/probably’, please use one or the other of the two words but not both.

FIGURES

Please consider avoiding the use of red and green in order to make your figure more accessible to those with colour blindness.

Figure 1: Please cite the reference numbers in square brackets as done in the main text. Citations should be preceding punctuation.

Figure 1: Please check the footnotes. The figure does not contain footnote ‘c’, but contains an asterisk which is not defined. Please define ‘HPV’, ‘FFPE’. For description of age, please add the unit ‘years’. 

Figure 2: Please add a unit for ‘age’.

Figure 3: Please add a unit for ‘age’. Please define ‘HPV’.

Figure 4: Please define ‘HPV’. Please clearly define the meaning of the dots and lines for the reader. Perhaps, ‘Impact numbers (dots) and 95% confidence intervals (lines)…’ in the title?

Figure 5: Please add a unit for ‘age’. Please define ‘HPV’. Please add a reference for the definition of the oncogenic types (as done in Table 1). Please clearly define the meaning of the dots and lines for the reader. Perhaps, ‘Impact numbers (dots) and 95% confidence intervals (lines)…’ in the title?

TABLES

Table 1: Please add a unit for ‘age’. Please define ‘HPV’. Please cite the reference numbers in square brackets.

Table 2: Please revise the number format (4254 versus 4,254). Please change ‘pseudo scenario’ to ‘pseudo-scenario’. Please add a unit for ‘age’. Please define ‘HPV’. In the footnote ‘e’, please write ‘age-standardized’ with a capital ‘A’. Where is footnote ‘b’ in the table?

Table 2: The definition of footnote ‘f’ might be misleading. We suggest: “Difference between the estimated number of cases in pseudo-scenario that all women were unscreened in last 10 years and the estimated number of cases in pseudo-scenario that all women were screened in last 10 years”.

Table 2: In footnote ‘g’, please introduce ‘CI’ as the abbreviation for ‘confidence intervals’.

Table 3: For clarity, we suggest exchanging the asterisk (*) and the slash (/) with ‘multiplied by’ and ‘divided by’. Please revise the number format (4254 versus 4,254). Please define ‘HPV’. Please add a unit for ‘age’. Please define ‘+∞’. 

Table 4: Please revise the number format (4254 versus 4,254). Please define ‘HPV’. Please add a unit for ‘age’. Please add a reference for the definition of the oncogenic types (as done in Table 1).

SUPPLEMENTARY MATERIAL

S1 Appendix: Please add a unit for ‘age’. Please define ‘HPV’ at first use. Please replace the word ‘cited’ with ‘Accessed’. Is the 1st ref a web ref? If so, please include the access date.

S3 Table: Please define ‘CL’ and the meaning of ‘+∞’. Please revise the number format (4254 versus 4,254). 

S3 Table A: Please add a definition for ‘all ages’. Could the CIs be presented in one column and separated by a comma? 

S3 Table B: Please add a unit for ‘age’. Please add a reference for the definition of the oncogenic types (as done in Table 1).

REFERENCES

Please revise reference 33 regarding the abbreviated journal title. Please ensure that journal name abbreviations match those found in the National Center for Biotechnology Information (NCBI) databases (http://www.ncbi.nlm.nih.gov/nlmcatalog/journals) and are appropriately formatted and capitalised.

SOCIAL MEDIA

To help us extend the reach of your research, please provide any Twitter handle(s) that would be appropriate to tag, including your own, your coauthors’, your institution, funder, or lab. Please respond to this email with any handles you wish to be included when we tweet this paper.

Comments from Reviewers:

Reviewer #1: The authors have adequately responded to my comment, and I have no additional observations.

Reviewer #2: The authors improved the manuscript that is much clearer now.

There is still an issue that I cannot agree and that should be discussed at least: the interpretation of the type-specific NNS. 

The authors answered to my comment #4 as follows: "When designing a screening program, it is essential to know if the condition screened for is rare or common. Which conclusion to make is up to the program - we are simply providing data that can be considered." I do not think this sentence applies to single genotypes. In fact, we start from HPV16 that is the most common and the most oncogenic type, then the decision, for the first level test of screening, is if we want to screen for HPV16 alone or HPV16 + the second most oncogenic type (let's say HPV18), therefore the condition HPV16+HPV18 will be more common than HPV16 alone, and so on. Single genotypes with high NNS will never be rare as condition to be screened, because they will be always considered as add on to a pool of other genotypes. Therefore, the NNF will determine if they have enough risk to be included in an efficient screening. NNS may give an idea of the public health impact of including a genotype or not, but cannot help to make the most efficient selection of types. 

In the discussion, I suggest to revise the sentence "If the NNS for a particular HPV type is high, meaning that the impact of the type to the screening program is low, consideration could be given as to whether screening could be discontinued for that type" (lines 318-320). This is surely true for a high NNF, but for a NNS the considerations are different: a high NNS may give a low value if we have to choice between a commercial test including that type and another not including it, but this is a different situation from deciding to dismiss screening for a single type. 

Reviewer #3: Alex McConnachie, Statistical Review

The authors have addressed all of my (minor) comments, and I have nothing else to add.

[LINK]

---

## [Editor Report · Decision Letter 3]

2 Oct 2023

Dear Dr Dillner, 

On behalf of my colleagues and the Academic Editor, Nicola Low, I am pleased to inform you that we have agreed to publish your manuscript "Impact of cervical screening by Human Papillomavirus genotype: Population-based estimations" (PMEDICINE-D-23-00933R3) in PLOS Medicine.

Thank you for your thoughtful and detailed responses to the editorial comments. We are pleased that the next step will be to publish your manuscript, but there are some minor stylistic and presentation aspects that should be addressed prior to publication. We will carefully check that these changes have been made. If there are any questions concerning these final requests, please do not hesitate to contact me at aschaefer@plos.org. 

Please see below for further minor points that we request you respond to:

1) Academic Editor comment: The text of footnote c in Figure 1 doesn't match what is in the box – it is difficult to match up the populations of women screened (656,607, 2012-2019) in the box with the legend, which talks about women in Stockholm 2012-2016. A reader cannot follow the numbers easily.

2) Abstract: Please change ll.38-39 to: “The primary limitation of our study is that the NNS is dependent on the HPV prevalence that can differ between populations and over time.

3) l.414: Please provide a reference for “largely related to HPV18”.

4) References: Please exchange ‘cited’ with ‘accessed’ when stating when websites were accessed (reference 15).

5) Figure 3: Please adjust the colors of the figure legend according to the colors used in the figure and explain in the figure description that the shaded areas represent the 95% confidence intervals.

PRESS

Sincerely, 

Alexandra Schaefer, PhD 

Associate Editor 

PLOS Medicine